# Higher measured than modeled ozone production at increased $NO_x$ levels in the Colorado Front Range

Bianca Baier[1,a,*], William Brune[1], David Miller[1], Donald Blake[2], Russell Long[3], Armin Wisthaler[4,5], Christopher Cantrell[6], Alan Fried[7], Brian Heikes[8], Steven Brown[9,10], Erin McDuffie[9,10,11], Frank Flocke[12], Eric Apel[12], Lisa Kaser[12], and Andrew Weinheimer[12]

[1]Department of Meteorology and Atmospheric Science, The Pennsylvania State University, University Park, PA, USA
[2]School of Physical Sciences, University of California, Irvine, CA, USA
[3]US EPA National Exposure Research Lab, Research Triangle Park, NC, USA
[4]Institute of Ion Physics and Applied Physics, University of Innsbruck, Austria
[5]Department of Chemistry, University of Oslo, Norway
[6]Department of Atmospheric and Oceanic Sciences, University of Colorado Boulder, Boulder, CO, USA
[7]INSTAAR, University of Colorado Boulder, Boulder, CO, USA
[8]Graduate School of Oceanography, The University of Rhode Island, Kingston, RI, USA
[9]Chemical Sciences Division, NOAA Earth System Research Laboratory, Boulder, CO, USA
[10]Department of Chemistry and Biochemistry, University of Colorado Boulder, Boulder, CO, USA
[11]Cooperative Institute for Research in the Environmental Sciences, University of Colorado Boulder, Boulder, CO, USA
[12]Atmospheric Chemistry Observations and Modeling Laboratory, National Center for Atmospheric Research, Boulder, CO, USA
[a]now at: Cooperative Institute for Research in the Environmental Sciences, University of Colorado Boulder, Boulder, CO, USA and Global Monitoring Division, NOAA Earth System Research Laboratory, Boulder, CO, USA

*Correspondence to:* Bianca Baier (bianca.baier@noaa.gov)

**Abstract.** Chemical models must correctly calculate the ozone formation rate, $P(O_3)$, to accurately predict ozone levels and to test mitigation strategies. However, air quality models can have large uncertainties in $P(O_3)$ calculations which can create uncertainties in ozone forecasts, especially during the summertime when $P(O_3)$ is high. One way to test mechanisms is to compare modeled $P(O_3)$ to direct measurements. During summer 2014, the Measurement of Ozone Production Sensor (MOPS) directly measured net $P(O_3)$ in Golden, CO, approximately 25 km west of Denver along the Colorado Front Range. Net $P(O_3)$ was compared to rates calculated by a photochemical box model that was constrained by measurements of other chemical species and that used a lumped chemical mechanism and a more explicit one. Median observed $P(O_3)$ was up to a factor of two higher than that modeled during early morning hours when nitric oxide (NO) levels were high and was similar to modeled for the rest of the day. While all interferences and offsets in this new method are not fully understood, simulations of these possible uncertainties cannot explain the observed $P(O_3)$ behavior. The $P(O_3)$ and peroxy radical ($HO_2$ and $RO_2$) discrepancies observed here are similar to those presented in prior studies. While a missing atmospheric organic peroxy radical source from volatile organic compounds co-emitted with NO could be one plausible solution to the $P(O_3)$ discrepancy, such a source has not been identified and does not fully explain the peroxy radical model-data mismatch. If the MOPS accurately depicts atmospheric $P(O_3)$, then these results would imply that $P(O_3)$ in Golden, CO would be $NO_x$-sensitive for more of the day than what is calculated by models, extending the $NO_x$-sensitive $P(O_3)$ regime from the afternoon further into the morning. These results

could affect ozone reduction strategies for the region surrounding Golden and possibly other areas that are in non-attainment with national ozone regulations. Thus, it is important to continue the development of this direct ozone measurement technique to understand P(O$_3$), especially under high NO$_x$ regimes.

## 1 Introduction

Ground-level ozone (O$_3$) is a hazardous air pollutant abundant in cities and their surrounding areas. Awareness of its detrimental health effects on both humans and plants has led to the Clean Air Act of 1970 and the development of National Ambient Air Quality Standards (NAAQS) (Krupa and Manning, 1988; Bell et al., 2004; US EPA, 2013, 2016b). Air pollution regulatory policies based on these standards have been successful in reducing O$_3$ by approximately 32% in the United States since 1980. However, current O$_3$ levels are stabilizing and even increasing again in the western United States (US EPA, 2016a). Understanding why these trends are occurring in areas despite more stringent emissions controls is crucial for further reduction of O$_3$ levels within the United States.

Boundary layer O$_3$ levels are dependent upon both chemical and meteorological processes described in the following equation:

$$\frac{\partial [O_3]}{\partial t} = P(O_3) + \frac{w_e \triangle O_3 - u_d [O_3]}{H} - \triangledown \boldsymbol{.} (\mathbf{v}[O_3]), \tag{1}$$

where $\partial [O_3]/\partial t$ is the local O$_3$ time rate of change, $P(O_3)$ is the instantaneous net photochemical O$_3$ production rate, $(w_e \triangle O_3 - u_d [O_3])/H$ is the combined entrainment and deposition rate of O$_3$ in or out of the mixing layer of height $H$, and $\triangledown \boldsymbol{.} (\mathbf{v}[O_3])$ is the O$_3$ advection rate. All of the physics, chemistry, and meteorology needed to solve this equation are included in chemical transport models (CTMs), which are used to design and test reduction strategies. For areas where local production is the dominant source of O$_3$, the term in Eq. (1) that will reduce O$_3$ through local emissions control strategies is P(O$_3$). Thus, understanding and accurately calculating O$_3$ formation is crucial for its mitigation.

Ozone formation chemistry has been well-documented for decades (Haagen-Smit et al., 1953; Finlayson-Pitts and Pitts, 1977; Seinfeld and Pandis, 2012; Calvert et al., 2015). The oxidation of volatile organic compounds (VOCs) by the hydroxyl radical (OH) produces hydroperoxy (HO$_2$) and organic peroxy (RO$_2$) radicals. These peroxy radicals react with nitrogen oxide (NO) to form nitrogen dioxide (NO$_2$), which is photolyzed to form new O$_3$ outside of the NO$_x$ photostationary state (PSS): a steady-state reaction sequence involving NO$_x$ (NO$_2$ + NO) and O$_3$. Thus, chemical O$_3$ production occurs through reactions with NO and peroxy radicals described in Eq. (2), where $k$ denotes a bimolecular reaction rate coefficient. Equation (3) describes the chemical O$_3$ (or NO$_2$) destruction rate or rate of removal to reservoir species as the fraction of O($^1$D) molecules resulting from O$_3$ photolysis that react with H$_2$O to form OH; reactions of O$_3$ with HO$_x$ (HO$_2$ + OH); the net production of peroxyacyl nitrates (PANs); the reaction of OH and NO$_2$ to form nitric acid (HNO$_3$); and O$_3$ loss through reactions with

alkenes and halogens. The net instantaneous $O_3$ production rate, $P(O_3)$, is then defined as the difference between $O_3$ chemical production and loss rates in Eq. (4):

$$P_{chem} = k_{NO+HO_2}[NO][HO_2] + \sum_{i=1}^{N} k_{NO+RO_{2i}}[NO][RO_2]_i \qquad (2)$$

$$L_{chem} = J_{O_3}f_{H_2O}[O_3] + k_{OH+O_3}[OH][O_3] + k_{HO_2+O_3}[HO_2][O_3] +$$
$$P(PANs) + k_{OH+NO_2}[OH][NO_2] + L(O_3)_{alkenes} + L(O_3)_{halogens} \qquad (3)$$

$$P(O_3) = P_{chem} - L_{chem}. \qquad (4)$$

Equations (2) and (3) illustrate the non-linear dependence of $P(O_3)$ on both $NO_x$ and the production of $HO_x$ from VOC oxidation. That is, these chemical species are involved in both the production and destruction of $O_3$ molecules. According to the current understanding, increases in NO can cause $P(O_3)$ to initially increase until $NO_x$ levels are sufficiently high to react with OH, thereby removing $HO_x$ and $NO_x$ from the reaction system and decreasing $P(O_3)$. Therefore, $P(O_3)$ is largely
dependent upon the cycling between $HO_x$ and $NO_x$ in the atmosphere; the exact $NO_x$ level at which this crossover occurs is sensitive to the production rate of $HO_x$ radicals (Jaegle et al., 1998; Trainer et al., 2000; Thornton et al., 2002; Ren et al., 2005). In a $NO_x$-sensitive regime, $P(O_3)$ varies with the square root of $P(HO_x)$ and decreases in $NO_x$ are more effective in decreasing $O_3$ than decreases in VOCs. Conversely, in a VOC-sensitive regime, $P(O_3)$ varies linearly with $P(HO_x)$ and decreases in VOCs are more effective in decreasing $O_3$, while further $NO_x$ decreases can act to increase $O_3$ (Kleinman et al., 1997; Seinfeld and
Pandis, 2012). Therefore, if the sensitivity of $P(O_3)$ to $NO_x$ and VOCs is known, efficient $O_3$ mitigation strategies can be devised that target precursor emissions and more effectively reduce $O_3$ in polluted regions.

     The gas-phase chemical mechanisms used in CTMs rely on a number of model input parameters to calculate $P(O_3)$ such as measurements of inorganic and organic chemical species; temperature- and pressure-dependent reaction rates; photolysis frequencies; and product yields of reactions. As the chemical processes contributing to $O_3$ formation are vast, complex, and
not fully quantified, it is difficult to portray atmospheric reactions in their entirety. Thus, mechanisms are simplified to describe the complex chemical state of the atmosphere. While inorganic chemistry is generally similar between reduced and more explicit mechanisms, differences in VOC aggregation schemes can create variance in modeled $P(O_3)$, $O_3$, or other important $O_3$ precursor predictions (Jeffries and Tonnesen, 1994; Olson et al., 1997; Kuhn et al., 1998; Luecken et al., 1999; Dodge, 2000; Tonnesen and Dennis, 2000; Jimenez et al., 2003; Luecken et al., 2008; Chen et al., 2010).
The current understanding of $O_3$ production chemistry is not consistent with all observations. The production of $O_3$ is dictated by reactions between peroxy radicals ($HO_2$ and $RO_2$) and NO, but prior studies have shown that measured and modeled

peroxy radicals are not always in agreement. Stone et al. (2012) provide a synthesis of model-measurement $HO_2$ comparisons, noting that model agreement with measurements is variable in low-$NO_x$ environments, but that models tend to underpredict $HO_2$ in urban, high-$NO_x$ areas. Such studies in high-$NO_x$ environments have shown that both zero-dimensional and three-dimensional modeled $HO_2$ – or the $HO_2$ to OH ratio – can be underestimated by up to a factor of ten at values of NO greater than a few parts per billion (Faloona et al., 2000; Martinez et al., 2003; Ren et al., 2003; Emmerson et al., 2005; Shirley et al., 2006; Emmerson et al., 2007; Kanaya et al., 2007, 2008; Dusanter et al., 2009; Chen et al., 2010; Sheehy et al., 2010; Ren et al., 2013; Czader et al., 2013; Brune et al., 2015; Griffith et al., 2016). For some studies, the maximum NO values were approximately 6 ppbv (Tan et al., 2017), so that the amount of model underestimation at high $NO_x$ values was within measurement uncertainty. Other studies show good agreement between model and measured $HO_2$ in the morning with average diel NO less than 2 ppbv (Hofzumahaus et al., 2009), or indicate good average agreement between measured and modeled $HO_2$, but indicate morning model $HO_2$ underestimation on individual days (Lu et al., 2013). In 2010, an interference involving partial conversion of $RO_2$ to $HO_2$ was found in $HO_2$ measurements that use reagent NO to convert $HO_2$ to OH, so that measured $HO_2$ was overestimated (Fuchs et al., 2011). Since the publication of that discovery, instruments are operated in a way that makes the interference negligible. However, even for measurements prior to 2011, atmospheric $HO_2/RO_2$ ratios suggest that the magnitude of an $HO_2$ interference likely accounts for no more than a factor of two in the difference between measured and modeled $HO_2$ (Cantrell et al., 2003). Removing this interference therefore improves model-measurement $HO_2$ agreement at high $NO_x$ within uncertainty levels in some studies, especially for NO less than 10 ppbv (Tan et al., 2017; Shirley et al., 2006), but cannot fully explain model $HO_2$ under-predictions at high $NO_x$ in others (Griffith et al., 2016; Brune et al., 2015).

Modeled organic peroxy radicals are also underestimated relative to measurements by up to a factor of ten (Hornbrook et al., 2011; Tan et al., 2017). Tan et al. (2017) show that this underestimation can be most prominent at high $NO_x$ and that further increasing a source of morning $RO_2$ proportional to this discrepancy improves agreement between measured and modeled peroxy radicals, but consequently overpredicts $HO_x$ species or OH reactivity. Studies in which $RO_2$ (and/or $HO_2$) are underestimated via model calculations at high $NO_x$ have examined possible missing VOCs or additional mechanisms that could reconcile this effect, but no such VOC or mechanism has been identified (Martinez et al., 2003; Kanaya et al., 2007; Dusanter et al., 2009; Hornbrook et al., 2011; Lu et al., 2012, 2013). To date, model under-prediction of peroxy radicals (either $HO_2$ or $RO_2$, and sometimes both species) at high $NO_x$ levels remains unresolved.

Due to the aforementioned discrepancies between measured and modeled radicals, $P(O_3)$ calculated from measured peroxy radicals can routinely be more than double the $P(O_3)$ calculated from modeled $HO_2$ or $RO_2$ at high $NO_x$, according to several field studies conducted during the past decade (Ren et al., 2003; Kanaya et al., 2008; Spencer et al., 2009; Ren et al., 2013; Brune et al., 2015; Griffith et al., 2016; Tan et al., 2017). The Measurement of Ozone Production Sensor (MOPS) directly measures $P(O_3)$ and can help to evaluate $O_3$ formation calculated from chemical mechanisms via Eqs. 2-4 (Cazorla and Brune, 2010; Baier et al., 2015). However, observed $P(O_3)$ has also shown similar discrepancies to modeled $P(O_3)$ at high NO or $NO_x$. For example, in 2010 the first version of the MOPS (Cazorla et al., 2012; Ren et al., 2013) compared directly-measured $O_3$ production rates to both modeled $P(O_3)$ and that calculated from measured peroxy radicals. Observed $P(O_3)$ and $P(O_3)$

calculated from measured radicals were approximately equal to that modeled for NO levels up to 1 ppbv, but were significantly larger for higher values of NO.

Inaccurate model P(O$_3$) estimation can directly affect O$_3$ forecasts. Im et al. (2015) and Appel et al. (2007) found that CTMs can underestimate O$_3$ levels above 60-80 ppbv and overestimate O$_3$ below 30 ppbv: errors that are typically credited to emissions and chemical mechanism choice. In addition, summertime O$_3$ predictions were most sensitive to regional production due to increased photochemical activity rather than transport (Im et al., 2015). It has also been found in the northeastern United States, that CTMs underestimated the effects of NO$_x$ emissions reductions on O$_3$ (Gilliland et al., 2008). Thus, details of the chosen chemical mechanism can greatly affect O$_3$ predictions and even reverse the order of O$_3$ production sensitivity to its precursors, decreasing confidence in models used for developing emissions reduction strategies.

P(O$_3$) was measured in Golden, CO in summer 2014 during a field study consisting of the Deriving Information on Surface conditions from COlumn and VErtically-resolved observations Relevant to Air Quality (DISCOVER-AQ) field campaign and the Front Range Air Pollution and Photochemistry Experiment (FRAPPÈ). This work describes comparisons between P(O$_3$) measured in situ by a second-generation MOPS and P(O$_3$) modeled using both lumped and near-explicit chemical mechanisms and we investigate the possible causes for differences observed between measured and modeled P(O$_3$).

## 2  Methods

### 2.1  MOPS measurements

A second-generation MOPS directly measures the instantaneous O$_3$ production rate, P(O$_3$), with an improved chamber and airflow design. The method is briefly described here; a more technical description of the MOPS and its modifications is detailed in Baier et al. (2015). The second-generation design aims to decrease artificial chemistry induced by air-surface interactions within the chambers. The difference in O$_x$ (defined here as O$_3$ + NO$_2$) is continuously sampled by two 26.9-L trapezoidal environmental chambers with airflow somewhat like a sheath flow to isolate sampled air from chamber surfaces. The sample chamber is transparent and undergoes the same O$_3$ photochemistry as the atmosphere, while the reference chamber is covered with a film that blocks all ultraviolet (UV) radiation of wavelengths below 400 nm, suppressing the radical chemistry essential for new O$_3$ production. Positioned after the chambers, a highly-efficient UV light-emitting diode photolyzes NO$_2$ into O$_3$ in air coming through separate tubing from both the sample and reference chambers. This converter cancels any differences in the NO$_x$ PSS caused by the reference chamber film. The difference in O$_x$, divided by the exposure time of air in the MOPS chambers, yields the net O$_3$ production rate as P(O$_x$).

The residence time is determined by adding a pulse of O$_3$ to the chambers and then measuring the O$_3$ as a function of time (Baier et al., 2015). The resulting pulse has a mean residence time of $130 \pm 5$s and the time at which the signal recedes into the background is 345 s. Thus, the exposure time of molecules in the chambers is taken to be 130 s.

The MOPS absolute uncertainty (1$\sigma$) is $\pm$ 11 ppbv h$^{-1}$ for 10-min measurements (Baier et al., 2015), but when averaged to one hour, this uncertainty decreases to approximately 5 ppbv h$^{-1}$. As previously mentioned in Baier et al. (2015), the MOPS technique can produce artificial negative O$_3$ production rates that appear to be roughly correlated with temperature, relative

humidity, or actinic flux. As discussed in Cazorla et al. (2012), negative $P(O_3)$ rates are unrealistic during the day when OH production is large enough to sustain new $NO_2$ and subsequent $O_3$ formation from VOC oxidation.

MOPS chamber loss tests and flow visualizations have been conducted to address these artifacts. Laboratory testing indicates that wall loss of $O_x$ and radical species is minimal (Cazorla and Brune, 2010; Baier et al., 2015). On the other hand, previous studies have found that commercial $O_3$ analyzers can exhibit both positive and negative responses to changes in relative humidity due to increases or decreases in water vapor (US EPA, 1999; Wilson and Birks, 2006). Additional laboratory testing has been conducted to investigate the sensitivity of the MOPS Thermo Scientific $O_3$ analyzer used in this study. Although differences in temperature between sample and reference chamber did not play a large role in initiating baseline drifting, the MOPS $O_3$ analyzer exhibits a large baseline shift greater than approximately 2 ppbv when air entering the analyzer has a relative humidity that is greater than 70%. This relative humidity threshold was determined by performing a laboratory simulation of field operations of the $O_3$ analyzer in its air-conditioned container.

The MOPS precision is typically 5 ppbv $h^{-1}$ ($1\sigma$) for one-hour averages, but $O_3$ analyzer drifting can degrade this precision. We quantify the MOPS diurnal $O_3$ analyzer drift and provide a correction to the raw $P(O_3)$ data through zeroing of the MOPS chambers, either by removing the reference chamber film for an entire day, or by measuring $P(O_3)$ on cooler, cloudy days when $O_3$ formation is likely low (Baier et al., 2015). On these occasions, the negative $O_3$ differential due to high relative humidity is apparent. Since the same $O_3$ formation will occur in both chambers on zero days, this method retrieves a baseline $P(O_3)$ time series that can be subtracted off of the MOPS raw data. Four zeros were applied to the raw $P(O_3)$ data during this study for an entire 24-hour period, with two using low $O_3$ production days as zeros. Positive deviations in the MOPS raw $P(O_3)$ from this negative baseline are evident during the morning hours, therefore this method extracts the positive $P(O_3)$ deviations from background $O_3$ analyzer drift during $O_3$ production hours of the day. Zeros that were taken only on days with diurnal patterns and absolute values of relative humidity that reflect the range of relative humidity measured on non-zeroing, MOPS measurements days were used in this analysis. Days chosen for zeroing the MOPS instrument with elevated ambient relative humidity compared to non-zeroing days likely have enhanced ozone analyzer drifting, according to our laboratory simulations of field operations. Thus, these days will not provide an average zero correction to the MOPS data because they are anomalous. The average zero correction that is subtracted from the raw $P(O_3)$ measurements to derive a corrected MOPS $P(O_3)$ is shown in Fig. S1. In addition, we have restricted our analysis to days when the MOPS ozone analyzer relative humidity was below 70% and have tested the robustness of this threshold using a wide range of MOPS relative humidities from 70%-90% to ensure that our corrected $P(O_3)$ values were not sensitive to this threshold choice (Fig. S2).

The MOPS chamber "sheath" airflow inhibits air that has contacted the walls from being sampled in the center of the chamber exit (Baier et al., 2015). Air is then sampled from a center flow that is isolated from the chamber walls. Smoke visualizations of the chamber flow, along with laboratory and atmospheric observations of chamber $O_x$ losses less than or equal to 5%, suggest that off-gassing of $O_x$ or other species from the MOPS chamber walls is inhibited and thus likely plays a negligible role in larger measured-than-modeled $P(O_3)$. From the laboratory and chamber testing to date, insignificant amounts of NO are lost in the MOPS chambers (Cazorla and Brune, 2010; Baier et al., 2015).

It is known that $NO_2$ adsorption onto the chamber walls can result in the heterogeneous formation of nitrous acid (HONO) through the reaction of $NO_2$ with water vapor adsorbed onto surfaces (Finlayson-Pitts et al., 2003); a photolytic HONO source has also been previously reported (Rohrer et al., 2005; George et al., 2005; Stemmler et al., 2006; Langridge et al., 2009; Lee et al., 2016; Crilley et al., 2016). During the 2013 DISCOVER-AQ study in Houston, TX, excess HONO of up to five times ambient values was measured in the MOPS chambers, which can thus create excess OH and positively bias the MOPS $P(O_3)$. Although a production mechanism has not been identified, this bias was found to be a) largest between 1000-1400 LT when $NO_x$ values are high, and b) correlated with relative humidity, temperature, and $J_{NO_2}$ (Baier et al., 2015). The MOPS inlet is one area that is suspected to facilitate HONO production due to inevitable surface interactions with sample air entering the chambers, and we have since decreased this bias by approximately 30% through shielding the MOPS inlet face and suppressing additional HONO production. Using the assumption that chamber HONO is generated from $NO_2$ adsorption on the chamber walls, chamber-generated HONO levels measured in Houston, TX were applied to this Golden, CO study by scaling the observed chamber HONO by ambient $NO_x$ levels; then the possible $P(O_3)$ interference due to chamber HONO was determined.

## 2.2 Site description and ancillary measurements

Second-generation MOPS measurements were recorded for 19 days in Golden, CO ($39^o44.623$'N, $105^o10.679$'W), which is located approximately 25 km west of the Denver metropolitan area. Commerce City, which houses several oil refineries, is located 30 km to the northeast. The Golden measurement site lies east of the Front Range, atop the South Table Mountain mesa (1833 m asl) and amidst grass-covered terrain. The Colorado summertime climate is hot and arid with intense solar radiation. These meteorological conditions can be conducive for high $O_3$ formation from both local and advected precursor emissions. Ozone production can also be affected by diurnally varying, thermally driven winds; morning heating of mountains invokes easterly upslope flow, transporting precursors from Denver and the urban corridor of the Front Range westward, while downsloping afternoon westerlies can re-circulate these pollutants eastward to lower elevations (Banta, 1984).

Measurements used to constrain the models in this study were obtained on ground-based and aircraft platforms during DISCOVER-AQ and FRAPPÈ. Both studies were co-located in the Colorado Front Range between 17 July and 10 August 2014. Continuous, ground-based 1-min measurements of meteorological parameters and inorganic chemical species include temperature, pressure, and relative humidity, $O_3$, sulfur dioxide, and $NO_x$. In the absence of continuous ground-based VOC measurements, $C_2$-$C_{10}$ non-methane hydrocarbons (NMHC) and organic nitrates were measured from 72 total whole-air canister (WAC) samples that were collected in Golden and analyzed by gas chromatography (GC) and gas chromatography-mass spectrometry (GC-MS) in the laboratory. An average of five samples were taken daily over 16 days. Approximately 64% of whole air sampling occurred between 0700 and 1200 local time (LT) to capture VOC mixing ratios during morning $O_3$ production hours, with sparser sampling in the afternoon between 1400 and 1800 LT to examine advection from sources east of Golden, CO such as the Denver metropolitan and Commerce City regions. Median diurnal values of VOCs were constructed from these point measurements to provide constraints for the model calculations. We initialized backward Hybrid Single-Particle Lagrangian Integrated Trajectory (HYSPLIT) models at 300 and 500 m heights beginning at 1600 LT and run for

12 hours using North American Model (NAM) meteorological data to determine whether the airflow in Golden could have originated from these eastern regions (Stein et al., 2015; Rolph, 2016). In general, higher $NO_x$ and anthropogenic VOC mixing ratios were measured when HYSPLIT indicated flow from these eastern pollution sources. Thus, for days when measurements were made in these plumes, separate median diurnal VOC values were constructed to more accurately represent the VOC speciation observed in Golden.

Canister VOCs were supplemented by boundary layer inorganic and organic chemical species measurements obtained on the NASA P-3B and NSF/NCAR C-130 aircraft and constant, median values were calculated for the limited times of the day when these aircraft were in the vicinity of Golden and used in the model (Table 1, Table S1). Aircraft measurements for Golden were available after 0900 LT on P-3B overflights which occurred up to three times daily, while C-130 measurements were available after 1000 LT when this aircraft was within roughly 20 km of the measurement site. Airborne measurements of inorganic and organic species agree to within 30% on average. More information on the DISCOVER-AQ and FRAPPÈ campaigns, aircraft and ground-based platforms, and measurement methods can be found at http://www-air.larc.nasa.gov/missions/discover-aq/discover-aq.html and https://www2.acom.ucar.edu/frappe.

### 2.3 Model description

Two types of chemical mechanisms were used in zero-dimensional photochemical box models to calculate $P(O_3)$ for the DISCOVER-AQ and FRAPPÈ campaign period. We used the lumped Regional Atmospheric Chemistry Mechanism version 2 (RACM2) (Stockwell et al., 1997; Goliff et al., 2013), and the near-explicit Master Chemical Mechanism version 3.3.1 (MCMv331) (Jenkin et al., 2003; Bloss et al., 2005; Jenkin et al., 2015). An exhaustive list of model constraints is displayed in Table 1. Cloud-free photolysis rates were calculated using the Tropospheric Ultraviolet (TUV) model (Madronich and Flocke, 1999) for Golden, CO. These photolysis rates were scaled to $J_{NO_2}$ calculated from continuous pyranometer measurements (LI-COR, LI-200 series) using the relationship described in Trebs et al. (2009) and then were used to constrain the models. All model constraints were interpolated to a 10-minute time step and input into the model to calculate $P(O_3)$ for the campaign period. The system of differential equations generated from both chemical mechanisms was integrated for 24 hours to allow reactive intermediates to reach steady-state. In addition, a one-day integration time is calculated to be sufficient for radical concentrations and intermediate species to reach steady-state as a two-fold or even three-fold increase in this integration time period did not impact radical concentrations or the $P(O_3)$ results described below. Longer-lived constituents not constrained in the model were given a 24-hour lifetime to both prevent buildup of these chemical species and to roughly account for advection or dilution losses. Modeled $P(O_3)$ is largely insensitive to this loss rate. We note that although transport and entrainment processes can also influence $O_3$ levels, zero-dimensional model runs described here do not include these processes. Instead, we focus on net $P(O_3)$ calculated with Eqs. 2-4 using modeled output.

### 2.4 Model uncertainty assessment

In order to explain calculated $P(O_3)$ behavior relative to the MOPS during hours of the day when there is typically a shift from VOC- to $NO_x$-sensitive $P(O_3)$ regimes, we explore model sensitivity to various inorganic and organic chemical species,

reaction rates, product yields, and other model parameters outlined in supplementary material. For cases in which modeled $O_3$ sensitivity is near the transition between VOC-sensitive and $NO_x$-sensitive, model $P(O_3)$ uncertainty can mask the proper designation of $O_3$ $NO_x$-VOC sensitivity (Chen and Brune (2012) and references therein). Thus, understanding the uncertainty of modeled $P(O_3)$ to model inputs and parameters defines what can be said about modeled $O_3$ sensitivity to VOCs and $NO_x$.

### 2.4.1 RACM2

The RACM2 model includes 119 species and 363 reactions and is run using the FACSIMILE solver (Stockwell et al., 1997; Goliff et al., 2013). An explicit isoprene chemistry scheme has replaced the original RACM2 isoprene chemistry and is highlighted in Paulot et al. (2009) and Mao et al. (2013). As this mechanism aggregates VOCs based on their functional groups and OH reactivity, the RACM2 significantly reduces the number of model inputs and parameters over more explicit mechanisms that treat VOCs and their intermediate products separately.

Model uncertainty is traditionally evaluated through sensitivity analyses in order to identify inputs (observational data) and parameters (reaction rates and product yields) that create the most variance in a model output of interest. These inputs are hereby called "influential" inputs. The RACM2 model uncertainty is assessed through the use of a global sensitivity analysis for daylight hours between 0600 and 1800 LT.

A Random Sampling-High Dimensional Model Representation (RS-HDMR) analysis was performed, which varies hundreds of model constraints with relatively low computational expense (Rabitz and Alis, 1999; Li et al., 2006, 2010). The variance in modeled $P(O_3)$ due to changes in influential model constraints was calculated, with the $P(O_3)$ $1\sigma$ uncertainty derived as the total $P(O_3)$ standard deviation divided by its mean from time periods evaluated between 0600 and 1800 LT. The RS-HDMR technique used for the RACM2 model runs is detailed in Chen and Brune (2012) and Chen et al. (2012). An overview of model input uncertainties and a description of this global sensitivity analysis are presented in supplementary documentation.

### 2.4.2 MCMv331

The MCMv331 (Jenkin et al., 1997; Saunders et al., 2003; Bloss et al., 2005; Jenkin et al., 2015), is freely available at http://mcm.leeds.ac.uk/MCM and is run using a MATLAB framework described in Wolfe et al. (2016). This mechanism includes roughly 6,000 species and 17,000 reactions, treats VOCs and their intermediates separately, and uses explicit isoprene degradation chemistry described in Jenkin et al. (2015). Because of the large number of inputs in this near-explicit mechanism, the MCMv331 uncertainty was assessed through a local sensitivity approach. That is, inputs were set to their upper and lower uncertainty limits (at a $1\sigma$ confidence level) in a one-at-a-time fashion while all other constraints were held at their original values. Total MCMv331 uncertainty was calculated by adding in quadrature the upper and lower percent deviations in $P(O_3)$ due to perturbations in model constraints relative to the MCMv331 base case. Input and parameter groups that were varied to derive this uncertainty are described in supplemental documentation.

## 3 Results

### 3.1 Campaign observations and P(O$_3$) time series

Observed and modeled P(O$_3$) were compared between 17 July and 10 August 2014 in Golden (Fig. 1). From 17-27 July, the campaign was characterized by a warmer, drier period followed by a relatively cooler, wetter period until the end of the study. Daily O$_3$ mixing ratios typically peaked between 1300-1800 LT with a median value of 59 ppbv. Higher O$_3$ levels exceeding 80 ppbv were observed on 22, 28, and 29 July as well as 3 August. The highest O$_3$ levels were observed on 22 July with a maximum mixing ratio of approximately 90 ppbv.

Due to the terrain of the Front Range, the average diel wind direction during the campaign period was westerly before 0900 LT, easterly to northeasterly from 0900 to 1400 LT, and then westerly again after 1400 LT, with diel-averaged speeds ranging between 2-3.5 m s$^{-1}$. Thus, it is possible for P(O$_3$) in Golden to be influenced by pollutants advected from nearby eastern source regions during the mid-morning to early afternoon.

The corrected MOPS P(O$_3$) maxima were routinely higher than 10 ppbv h$^{-1}$ on most measurement days, with diurnal peaks between 0900-1100 LT. Observed P(O$_3$) maxima on individual days range from 10 ppbv h$^{-1}$ to almost 30 ppbv h$^{-1}$ (Fig. 1). As mentioned earlier, MOPS P(O$_3$) measurements were restricted to days when the MOPS O$_3$ analyzer relative humidity is less than 70% when we have confidence that the analyzer was not affected by significant baseline drifting. This data filtering reduced the MOPS baseline variations to between -5 and 5 ppbv h$^{-1}$ at a 1-hour time resolution.

### 3.2 Modeled P(O$_3$) time series and comparisons to measurements

Full-campaign modeled P(O$_3$) is also shown in Fig. 1 for both RACM2 and MCMv331. Modeled P(O$_3$) for both mechanisms are a broad peak with maxima that occurred between 0900-1200 LT with values generally 10 ppbv h$^{-1}$ or lower. The modeled P(O$_3$) behavior is essentially identical on a day-to-day basis for both the RACM2 and MCMv331. On several individual days, the MOPS P(O$_3$) measurements exhibited maxima that were a factor of two to three times higher than modeled P(O$_3$) values during the morning between 0900-1100 LT.

Median diel variations of MOPS and modeled P(O$_3$) are shown for MOPS measurement days in Fig. 2. Median observed P(O$_3$) began to increase around 0800 LT, peaked at greater than 10 ppbv h$^{-1}$ around 1000 LT, and decreased to 5 ppbv h$^{-1}$ before falling off to zero in the evening. Median modeled P(O$_3$) also rose beginning at about 0800 LT but peaked at around 5 ppbv h$^{-1}$ between 1100 and 1200 LT, and was 3-4 ppbv h$^{-1}$ in the afternoon. Median observed and modeled P(O$_3$) are in good agreement in the afternoon as shown by overlapping errorbars, but median diel MOPS P(O$_3$) is generally a factor of two higher than that modeled between 0900-1100 LT when NO$_x$ and VOC levels were high due to abundant local or advected rush hour traffic emissions. The shaded region in Fig. 2 is the range of possible measured P(O$_3$) values obtained using the range of maximum to minimum measured zero offset values. This mid-morning difference between measured and modeled diel-averaged P(O$_3$) is apparent over this range of zero corrections.

Figure 3 indicates P(O$_3$) as a function of NO levels and time of day. Similar to Cazorla et al. (2012), both measured and modeled diel P(O$_3$) increased between 0600-0800 LT during morning rush hour, peaked before 1200 LT, and then decreased

later in the day with decreasing NO and VOC radical abundances. Occasional secondary $P(O_3)$ peaks were exhibited between 1400-1600 LT in both measured and modeled $P(O_3)$, likely due to advection of $O_3$ precursors from the Denver region or increased local traffic emissions. The most striking difference is that the measured $P(O_3)$ continues to rise as NO increases while the modeled $P(O_3)$ decreases for NO more than 1 ppbv. The missing modeled $P(O_3)$ appears to increase monotonically with increasing NO for NO values greater than roughly 1 ppbv (Fig. 4). The difference between measured and modeled $P(O_3)$ is near zero up to 1 ppbv NO and almost 15 ppbv h$^{-1}$ at 5 ppbv NO. This unexpected increase in $P(O_3)$ with increasing NO provides a clue as to what might be causing the difference between measured and modeled $P(O_3)$.

Several reasons provide confidence in these $P(O_3)$ comparisons, which result in higher $P(O_3)$ than that modeled during the morning hours. First, median $P(O_3)$ values were used instead of the mean to compare MOPS and modeled $P(O_3)$ so as not to bias diurnal $P(O_3)$ curves high or low in the event of $P(O_3)$ anomalies. Second, observed $P(O_3)$ peak values were often much greater than the hourly MOPS $1\sigma$ uncertainty on individual days as seen in Fig.1, where differences between the MOPS and modeled $P(O_3)$ were typically between 10-20 ppbv h$^{-1}$. Third, when different relative humidity thresholds are used to correct the raw $P(O_3)$ data, measured $P(O_3)$ consistently exhibits the same diurnal behavior with a positive deviation from modeled $P(O_3)$ around 1000 LT. Fourth, deviations from the $O_3$ differential baseline derived from zeroing methods are observed between 0900-1100 LT even before correcting the MOPS measurements. Thus, we have confidence in the positive MOPS $P(O_3)$ signatures, which are greater than the modeled $P(O_3)$ during the morning hours. All of these results provide confidence in the robustness of the MOPS behavior relative to the models in Figs. 1 and 2 and in the subsequent analyses.

### 3.3 Possible causes of the model-measurement $P(O_3)$ discrepancies

Higher morning $P(O_3)$ calculated from measured peroxy radicals has been observed at high NO with a variety of measurement methods. The MOPS observations, independent of these studies, yield similar results for the dependence of $P(O_3)$ on NO indicating that the MOPS and other measurement methods both contain artifacts that act to increase $P(O_3)$ in a similar manner, or that the model-measurement disagreement occurs due to differences in the chemistry between observational and computational methods used to determine $O_3$ production rates.

We explore several hypotheses for model-measurement disagreement during the morning hours in the following sections. Possible explanations include MOPS chamber artifacts, model input and parameter uncertainties, model peroxy radical chemistry, modeled ambient HONO sources, and reactive chlorine chemistry.

#### 3.3.1 MOPS chamber artifacts

One hypothesis is that the MOPS $P(O_3)$ is positively biased due to environmental chamber chemistry artifacts: that is, offgassing of $NO_2$, nitrous acid (HONO), or other chemical species from the chamber walls. At higher relative humidity, chemical species adsorption onto these environmental chamber walls can be higher (Wainman et al., 2001). It is possible that subsequent desorption of $NO_2$ or chemical species from the walls can induce artificial chemistry in the MOPS chambers. However, as described earlier, the MOPS chamber airflow isolates sampled air from the walls of the MOPS chambers where surface reactions

are most likely to occur. Chamber air closest to the walls is exhausted, leaving mostly center flow to be sampled by the MOPS $O_3$ analyzer.

Additionally, adsorbed $NO_2$ can result in heterogeneous formation of HONO, and a HONO source within the chambers may result in excess $P(O_3)$ from artificial OH production (Baier et al., 2015). For the Golden, CO study, $NO_x$ levels were

a factor of three lower on average than in Houston, TX, the relative humidity was 35% lower on average, and the actinic flux was similar. Although identifying MOPS chamber HONO production mechanisms will require more intensive laboratory studies, we assume that the largest HONO source within the MOPS chambers stems from $NO_2$ adsorption on the chamber walls. Thus, $NO_x$ levels in Golden are used to infer MOPS chamber HONO levels in Golden. We have applied the observed chamber HONO:$NO_x$ ratio in Houston, TX to the Golden, CO study because, under this assumption, HONO production

should depend linearly on $NO_x$ adhering to the walls. We have calculated a maximum diurnal bias of +3 ppbv h$^{-1}$ at 1000 LT ($2\sigma$) that decreases later in the day to less than 1 ppbv h$^{-1}$ as $NO_x$ decreases. However, this calculated $P(O_3)$ bias is rather conservative; the chamber residence time of 130 s and the HONO photolysis frequency for Golden, CO can be used to determine the percentage of chamber HONO that would be converted into $O_3$-producing radicals. In doing so, less than 15% of chamber HONO is photolyzed. Consequently, the bias for Golden, CO would be less than 0.5 ppbv h$^{-1}$ and would

contribute insignificantly to the observed $P(O_3)$ signal. In order to explain observed and modeled $P(O_3)$ differences in Golden by chamber-induced HONO production, HONO levels would need to be more than an order of magnitude larger. Given the levels of MOPS chamber HONO measured in Houston and in other areas (Baier et al., 2015), the likelihood of excess chamber HONO production being a significant cause for the resultant differences between modeled and measured $P(O_3)$ is small.

### 3.3.2 Model input and parameter uncertainty

A second hypothesis is that the uncertainties in the model $P(O_3)$ are large enough that the differences in the measured and modeled $P(O_3)$ are not statistically different. Model $P(O_3)$ uncertainty has been found to be 2-5% larger during the morning hours when differences between measured and modeled $P(O_3)$ were observed. As described earlier, the RACM2 inputs and parameters affecting model $P(O_3)$ uncertainty are determined based on a RS-HDMR sensitivity analysis. Model uncertainty between 0600 and 1800 LT is similar between both chemical mechanisms (Table S4); the average modeled $P(O_3)$ uncertainty

($1\sigma$) from RACM2 and MCMv331 about 30% all day. Due to similar model behavior and diurnal uncertainty estimates between the RACM2 and the MCMv331, we expect that the influential inputs between the two mechanisms – model constraints and parameters contributing largely to calculated $P(O_3)$ uncertainty – will also be similar.

Model influential inputs are specific to both location and available measurements. However, many constraints that contributed to $P(O_3)$ uncertainty in Golden, CO were found to be similar to prior sensitivity analyses of chemical mechanisms

conducted in much different environments (Chen and Brune (2012) and references therein). For example, two parameters consistently identified as having high importance for daytime $P(O_3)$ uncertainty involve the reaction rates, k$_{OH+NO_2}$ and k$_{HO_2+NO}$, which dictate HO$_x$-NO$_x$ cycling and the production and loss of HO$_x$. These reaction rate coefficients have large contributions to the overall model uncertainty despite their relatively low uncertainty factors of 1.3 and 1.15 respectively (Sander et al., 2011).

Other model constraints influential in dictating model P(O$_3$) uncertainty such as reaction rates, product yields and mixing ratios of species were more specific to time of day. Similar to overall results in Chen and Brune (2012), and in addition to HO$_x$-NO$_x$ reaction rates, early morning P(O$_3$) uncertainty was attributed to reaction rates involving the oxidation of reactive VOCs such as aldehydes and xylenes that initiate O$_3$ formation propagation and produce HO$_x$. Additional Golden influential reaction rates involved the decomposition and formation rates of peroxyacyl nitrates (PAN), a NO$_x$ reservoir. As O$_3$ increases in the afternoon, additional rates and product yields of reactions involving O$_3$ loss also become important, along with those between NO and other organic peroxy species (RO$_2$) that continue O$_3$ formation.

As expected, model inputs and parameters involving the formation of RO$_2$ or new NO$_2$ outside of the NO$_x$ PSS that further propagate the O$_3$ formation cycle, along with inputs and parameters involving production of HO$_x$ species, are all factors influencing model P(O$_3$) uncertainty. Although model uncertainty is not large enough to explain model P(O$_3$) behavior relative to the MOPS, greater emphasis should be placed on quantifying the uncertainty in HO$_x$-NO$_x$ cycling reaction rates to reduce model P(O$_3$) uncertainty and improve morning agreement between observed and modeled P(O$_3$) in Figs. 1 and 2.

### 3.3.3 Model peroxy radical chemistry

An explanation for the lower modeled P(O$_3$) in early morning is that modeled HO$_2$ is less than that measured at higher NO. Indeed, in previous studies, measured HO$_2$ often exceeds modeled HO$_2$ for NO greater than about 1 ppbv (Faloona et al., 2000; Martinez et al., 2003; Ren et al., 2003; Shirley et al., 2006; Emmerson et al., 2007; Kanaya et al., 2007; Dusanter et al., 2009; Sheehy et al., 2010; Chen et al., 2010; Ren et al., 2013; Brune et al., 2015). Campaign median NO mixing ratios typically peaked between 0900-1100 LT at about 2 ppbv with maxima as high as 7 ppbv. The largest differences in measured and modeled P(O$_3$) occur during this time period when NO was greater than 1 ppbv. Thus, it is possible that the difference between measured and modeled HO$_2$ is related to the difference between measured and modeled P(O$_3$).

Measurements of HO$_2$, RO$_2$ (35% accuracy, 2$\sigma$) and OH (45% accuracy, 2$\sigma$) were made on board the NSF/NCAR C-130 using chemical ionization mass spectrometry (CIMS) (Mauldin et al., 2003; Hornbrook et al., 2011). Figure 5 indicates that the CIMS HO$_2$/OH ratio is approximately equal to the modeled HO$_2$/OH ratio for NO less than 1 ppbv but surpasses modeled HO$_2$/OH for NO greater than 1 ppbv, declining less rapidly than models for increasing NO mixing ratios. In previous studies, the agreement between measured and modeled OH has been independent of NO, so that the deviation between the measured and modeled HO$_2$/OH ratio is due to deviations between measured and modeled HO$_2$ (Shirley et al., 2006; Kanaya et al., 2007; Dusanter et al., 2009; Sheehy et al., 2010; Ren et al., 2013; Czader et al., 2013; Brune et al., 2015). Modeled RO$_2$ relative to the CIMS observed RO$_2$ is also underestimated at high NO (Fig. 5). Because the C-130 aircraft and ground-based inorganic and organic species mixing ratios in Golden are within 30% on average, a disagreement between measured and modeled peroxy radicals at high NO observed on the aircraft is relevant to understanding the MOPS measurements made at the Golden ground-based site.

One hypothesis is that a missing HO$_2$ or RO$_2$ source was not included in the models and that this source may be proportional to NO. However, forty-two total C$_2$-C$_{10}$ VOCs were measured by whole-air canister samples representing a large suite of organic chemical species within the models including ones with high OH reactivities that are particularly important for O$_3$

formation. We have tested two hypotheses: first, that an $RO_2$ source that reacts with NO to form $HO_2$ and $NO_2$ is missing in the models despite the suite of VOC measurements made in Golden, and second, that an unknown peroxy radical source co-emitted with NO can explain the missing modeled $P(O_3)$.

To test our first hypothesis, we have added a generic reaction involving $RO_2$ and NO to form additional $HO_2 + NO_2 + RO$ in the MCMv331, enhancing the $HO_2$ produced from $RO_2 + NO$ reactions. Since the reaction between the methylperoxy radical ($CH_3O_2$) and NO is the dominant organic peroxy radical reacting with NO to form new $O_3$ in both chemical mechanisms, this species' reaction rate coefficient was used for this model test. By essentially doubling the $CH_3O_2$ rate constant, this reaction only elevates modeled $P(O_3)$ throughout the day, does not alter the diurnal $P(O_3)$ pattern (Fig. 6), and does not resolve the discrepancy between measured and modeled peroxy radicals at high NO.

To test our second hypothesis, additional $RO_2$ was added to the model in order to match the peak morning diel MOPS $P(O_3)$ in Fig. 6, and was scaled to NO as this unknown species is likely co-emitted with NO. This addition improves model-measurement $P(O_3)$ agreement in the morning and only slightly overestimates the afternoon diel $P(O_3)$ relative to the MOPS, agreeing with measured $P(O_3)$ within uncertainty levels. In magnitude, the model-measurement $RO_2$ agreement is improved with this case study. However, the $RO_2$ continues to increase while measured $RO_2$ decreases at higher NO (Fig. 5). Thus, this model case study suggests that an unknown missing $RO_2$ source could possibly explain the differences between measured and modeled $P(O_3)$ if this discrepancy between measured and modeled $RO_2$ can be resolved and if the identity of the unknown $RO_2$ can be found.

A VOC source that could explain prior model-measurement $HO_x$ disagreement has not been identified in other literature studies where missing $HO_2$ was of similar magnitude to this study, even when proposed VOCs were added to model base case scenarios (Martinez et al., 2003; Kanaya et al., 2007; Dusanter et al., 2009). Brune et al. (2015) discuss that, if this missing $HO_x$ source is also a missing OH loss, then this loss would be evidenced in measurements of OH reactivity at high NO, yet no such OH loss was observed. Further, Spencer et al. (2009) found that measured peroxynitric acid ($HO_2NO_2$) is also elevated compared to models at high NO or $NO_x$. Peroxynitric acid thermally decomposes to form $HO_2$ and $NO_2$, and can also be weakly photolyzed to form $HO_2$. Kanaya et al. (2007) propose that increasing the thermal decomposition rate of $HO_2NO_2$ could resolve model underestimation of $HO_2$ at high NO, but even when this decomposition rate was increased by a factor of five, it did not correct for higher measured than modeled $P(O_3)$ at high NO. Model sensitivity runs for Golden, CO using this increased decomposition rate for $HO_2NO_2$ in MCMv331 corroborate this same result (Fig. 6).

### 3.3.4  Modeled ambient HONO sources

Another hypothesis is that ambient HONO is missing from the model. The production and subsequent photolysis of nitrous acid (HONO) is an important morning $HO_x$ source at high NO or $NO_x$, often comparable to or larger than other $HO_x$ sources such as peroxide and organic VOC photolysis or $O_3$ photolysis followed by the subsequent reaction between $O(^1D)$ and water vapor to produce OH. In previous field studies, HONO photolysis was a substantial contributor to daytime $HO_x$ production, but can be largely underpredicted by models, especially in urban environments and may be a more viable solution to the model-measurement discrepancy found in this study.

Nitrous acid was not measured during DISCOVER-AQ or FRAPPÈ, but was predicted by the gas-phase RACM2 and MCMv331 based on continuous, ground-based $NO_x$ observations. Model HONO sources in this study only include those in the gas phase. Photolytic conversion of $NO_2$ to HONO on aerosol surfaces (Kleffmann et al., 1998; Arens et al., 2001; Monge et al., 2010); adsorption of $HNO_3$ on ground surfaces and subsequent photolysis (Zhou et al., 2003, 2011) and other photolytic heterogeneous sources are not included. Therefore, model under-prediction of HONO mixing ratios in the morning can be one cause for modeled versus measured $HO_2$/OH disagreement.

Lee et al. (2016) indicate that, even after additional gas-phase and heterogeneous HONO sources were added to MCMv331, HONO was still underestimated relative to models on average, and that a missing HONO source was correlated with $J_{NO_2}$, $NO_2$, and the product of $NO_2$ and OH reactivity for an urban area. Furthermore, only model results using measured HONO were able to replicate observed OH levels (Lee et al., 2016). Field studies in which HONO was continuously measured and used to constrain both zero-dimensional and three-dimensional chemical models have been able to replicate observed OH within uncertainty levels, but still exhibit the same behavior of higher measured-than-modeled $HO_2$ to OH ratios and $P(O_3)$ at high NO (Ren et al., 2003; Martinez et al., 2003; Dusanter et al., 2009; Chen et al., 2010; Czader et al., 2013; Ren et al., 2013; Brune et al., 2015).

A HONO source proportional to $NO_x$ was added to the MCMv331, resulting in average HONO levels of 0.5-0.9 ppbv between 0700 and 1200 LT, with peak HONO levels of 0.9 ppbv at 1000 LT when MOPS $P(O_3)$ exhibits its diel peak. This case study approximately replicates the observed morning $P(O_3)$ (Fig. 6) and observed OH within uncertainty levels. However, while added HONO in the MCMv331 improves the agreement between observed and modeled diel $P(O_3)$, mid-morning HONO levels needed to do so are over a factor of two higher than those observed in other areas within Colorado (Brown et al., 2013; VandenBoer et al., 2013) and in environments with much higher $NO_x$ levels (VandenBoer et al., 2015). Thus, the $HO_2$/OH ratio and the abnormally high HONO required to match the observed $P(O_3)$ provide evidence that at most only a part of the observed $P(O_3)$ can be explained by atmospheric HONO.

### 3.3.5 Reactive chlorine chemistry

Model under-representation of nitryl chloride ($ClNO_2$) production is another possible cause of the model underestimation of $P(O_3)$. Nitryl chloride serves as a nocturnal $NO_x$ reservoir and, when photolyzed, can produce additional reactive chlorine (Cl) and nitrogen dioxide ($NO_2$). Reactive chlorine, even at low mixing ratios, has been found to serve as a major oxidant for VOCs, possibly increasing $HO_2$ and $O_3$ production in the early morning hours by as much as 30% (Finlayson-Pitts et al., 1989; Atkinson et al., 1999; Chang et al., 2002; Osthoff et al., 2008). The effects of $ClNO_2$ production on chlorine chemistry and VOC oxidation have been provided in the literature as one possible explanation for measured versus model-data $HO_2$ mismatch at higher NO (Thornton et al., 2010; Riedel et al., 2014; Xue et al., 2015).

Heterogeneous uptake of dinitrogen pentoxide ($N_2O_5$) on chloride-containing aerosol particles can produce nitric acid ($HNO_3$) and $ClNO_2$ in both marine and continental environments through the following reaction:

$$\text{N}_2\text{O}_5 \xrightarrow{\text{k}_{het}} \phi\,\text{ClNO}_2 + (2 - \phi)\text{HNO}_3, \hspace{4cm} \{1\}$$

where $k_{het}$ is the heterogeneous reaction rate coefficient dependent upon the aerosol surface area density and the $\text{N}_2\text{O}_5$ uptake coefficient on chloride-containing aerosols, and $\phi$ is the $\text{ClNO}_2$ product yield.

To test this hypothesis, we constrained the MCMv331 with continuous, cavity ring-down spectroscopy measurements of $\text{N}_2\text{O}_5$ (Brown et al., 2002) from a nearby measurement site (Boulder Atmospheric Observatory; $40.050^o$N, $105.010^o$W), and implemented a reduced chlorine chemical mechanism in the MCMv331 provided by Riedel et al. (2014). We assumed an $\text{N}_2\text{O}_5$ uptake coefficient of 0.02, which is within the range of coefficients calculated in prior field studies (Wagner et al., 2013; Riedel et al., 2013) and laboratory experiments (Zetzsch and Behnke, 1992; Behnke et al., 1997). To be consistent with previous studies near Golden, the aerosol surface area density was varied between 150 and 250 $\mu$m$^2$ cm$^{-3}$, and $\phi$ is varied

between 0.05 and 0.1 (Thornton et al., 2010; Riedel et al., 2013). It is important to note that these assumptions vary largely with relative humidity and aerosol surface area and composition (Thornton and Abbatt, 2005; Bertram and Thornton, 2009; Roberts et al., 2009; Thornton et al., 2010; Wagner et al., 2013; Riedel et al., 2013), but modeling over a range of values can provide a qualitative prediction of $\text{ClNO}_2$ production effects on model P($\text{O}_3$) in this region. In each model case, the MCMv331 runs including $\text{ClNO}_2$ production and Cl-VOC chemistry resulted in average $\text{ClNO}_2$ mixing ratios between 0.04 and 0.13 ppbv

during the early morning hours (0300-0600 LT) and a slight increase in diurnal P($\text{O}_3$) values of less than 5%. Thus, although chlorine chemistry can have a large effect on P($\text{O}_3$) during the winter and for marine environments, these model runs indicate that Cl chemistry does not play a large enough role in $\text{O}_3$ photochemistry during this summer campaign to explain the morning observed discrepancy between measured and modeled $\text{O}_3$ formation rates in Golden, CO.

### 3.4   Implications for $\text{O}_3$ mitigation strategies

### 3.4.1   $\text{NO}_x$-VOC sensitivity

The underestimation of model P($\text{O}_3$) relative to the MOPS at high NO or $\text{NO}_x$ can have far-reaching implications for model assessment of the dependency of P($\text{O}_3$) on $\text{NO}_x$ and VOCs. When examining model sensitivity to $\text{NO}_x$, levels were adjusted up or down by a factor of two and as a result, increasing $\text{NO}_x$ levels decreases P($\text{O}_3$) (as in a VOC-sensitive regime) while lowering $\text{NO}_x$ levels acts to increase P($\text{O}_3$) (Fig. 6).

The fraction of free radicals removed by $\text{NO}_x$, $L_N/Q$, has been used in the literature to assess $\text{NO}_x$-VOC sensitivity in regions experiencing high $\text{O}_3$ (Daum et al., 2004; Kleinman, 2005; Ren et al., 2013). Here, $L_N$ is the rate of total free radical removal by $\text{NO}_x$, and $Q$ is the total radical production rate. When significantly above 0.5, the atmosphere is within a VOC-sensitive regime, while when significantly below 0.5, the atmosphere is within a $\text{NO}_x$-sensitive regime (Kleinman, 2005). The median $\text{L}_N/Q$ was calculated with the RACM2 using full-campaign observations, indicating that the Golden, CO modeled

P($\text{O}_3$) is VOC-sensitive before 1200 LT and $\text{NO}_x$-sensitive thereafter (Fig. S4). During DISCOVER-AQ and FRAPPÈ, model

sensitivity studies conducted for the Boulder Atmospheric Observatory site just northeast of Golden also found maximum photochemical $O_3$ to be largely $NO_x$-sensitive in the afternoon (McDuffie et al., 2016). If peroxy radicals are underestimated by chemical mechanisms relative to observations for NO levels greater than a few ppbv, then the total radical production rate, Q, may also be underestimated, thereby shifting $L_N$/Q towards $NO_x$-sensitivity in the early morning and prolonging this regime
during times of the day when $O_3$ production is largest.

The largest $O_3$ formation rates are measured between 0900-1100 LT when $NO_x$ and VOC emissions are high and the mixing layer depth is relatively developed at 600 to 1000 m on average. Although a shallower mixing layer could be one reason for high MOPS P($O_3$) before 1100 LT, we note that secondary diurnal MOPS P($O_3$) peaks are also evidenced on individual days alongside increased $NO_x$ and VOCs during afternoon rush hour in a fully-developed mixing layer. Further, high P($O_3$) and the
shift from VOC- to $NO_x$-sensitivity in the late morning could be attributed to early-morning entrainment of VOCs from the free troposphere in the absence of $NO_x$ entrainment. However, these VOCs in the upper troposphere are longer-lived and are less important in propagating $O_3$ formation than other, higher reactivity VOCs. Therefore, although entrainment of species during the morning hours and the depth of the mixing layer influence $NO_x$-VOC sensitivity and these high morning P($O_3$) rates, it is more likely that $O_3$ precursor species at the surface level are the predominant factors influencing P($O_3$) for this study.
Although longer-term analyses are generally required to suggest effective $O_3$ reduction strategies, if the P($O_3$) $NO_x$-VOC sensitivity is shifted more towards a $NO_x$-sensitive regime in the morning as the MOPS observations suggest, reducing $NO_x$ would be an effective strategy for $O_3$ mitigation in Golden, CO and its immediate surroundings.

### 3.4.2   $O_x$ advection

Ozone formation precursors can be transported westward to Golden because of the Colorado Front Range terrain and its induced
wind patterns. When air in Golden is influenced by $O_3$ precursor emissions from the east (e.g. the Denver metropolitan and Commerce City regions), greater anthropogenic VOC and NO mixing ratios are measured on average. Thus, we evaluate calculated $O_3$ advection using Eq. (1) in an attempt to evaluate the impact of $O_3$ advection derived from the MOPS and the models on observed $O_3$ patterns in Golden.

Measured $O_x$ maxima are 2-7 ppbv greater on these "plume" days than when air is advected from elsewhere, and higher
P($O_3$) is measured by the MOPS than is modeled by the RACM2 and MCMv331 (Fig. 7). This result is roughly consistent with the difference between measured and modeled P($O_3$) as a function of NO shown in Fig. 4. When winds are not easterly ("non-plume" days), lower levels of anthropogenic VOCs and NO, and lower $O_x$ maxima are observed. Average measured diel P($O_3$) is also 20% lower than on plume days. The MOPS behavior stands in contrast to the models, where average diel RACM2 and MCMv331 P($O_3$) is approximately 30% higher on non-plume days than on plume days.
A simple advection analysis was performed to determine the factors in Eq. (1) that most contribute to observed $O_x$ levels in Fig. 7 for the campaign period. The transport rate of $O_x$ out of the mixing layer through deposition is calculated to be at most 1 ppbv h$^{-1}$ and is neglected here. The morning $O_3$ entrainment rate during DISCOVER-AQ and FRAPPÈ has been calculated for the Colorado Front Range region to be 5 ppbv h$^{-1}$ on average, with afternoon average entrainment rates of approximately -1 ppbv h$^{-1}$ (Kaser et al. (2016), *in prep*). Assuming an average entrainment rate of 5 ppbv h$^{-1}$ for morning hours between

0600-1200 LT and an $O_x$ entrainment rate of -1 ppbv h$^{-1}$ for times between 1200-1800 LT and subtracting diel entrainment and observed P($O_x$) from the local diel $O_x$ rate of change, the average $O_x$ advection rate derived from MOPS and models between 0600-1800 LT is -5.4 to -2.4 ppbv h$^{-1}$ on plume days, and -1.7 to -3.5 ppbv h$^{-1}$ for all other days, respectively. This quick calculation suggests that advection contributes weakly to observed $O_x$, while either entrainment or P($O_x$) dominate the

$O_x$ patterns observed in Golden and its surrounding areas. Because these advection rates are derived quantities from the MOPS and the models, and both methods for determining P($O_x$) contain substantial uncertainty, it is difficult to quantitatively assess $O_x$ advection rates in Golden, CO. Decreasing the uncertainty in P($O_x$) is thus salient for accurately calculating the terms in Eq. (1) contributing to observed $O_x$ levels in the Colorado Front Range.

## 4  Conclusions

Comparisons were made between P($O_3$) measured in situ by a second-generation Penn State MOPS and photochemical box modeled P($O_3$) using both lumped and near-explicit chemical mechanisms. These comparisons during the 2014 DISCOVER-AQ and FRAPPÈ field campaigns in the Colorado Front Range show that median diel modeled P($O_3$) is underestimated relative to the MOPS by roughly a factor of two in mid-morning, when actinic flux is increasing and morning rush hour abundances of NO$_x$ and VOCs are decreasing. This result corroborates to previous studies that had P($O_3$) measured by MOPSs (Cazorla

et al., 2012; Baier et al., 2015). Thus, this model-data P($O_3$) mismatch appears to come from unknowns in the chamber or from unknown peroxy radicals missing in the models and not from one particular environment.

The uncertainties in both the measurement and the model are substantial. The measurement uncertainty is about $\pm$ 5 ppbv h$^{-1}$ for one-hour measurements, with the largest portion due to the zeroing of the daily negative drifting of the differential $O_3$ measurement. Model P($O_3$) uncertainty is about 30% ($1\sigma$ confidence) during peak P($O_3$) hours; factors such as uncertainty

in the kinetic rate coefficients of HO$_x$-NO$_x$ cycling reactions are most significant. Despite these uncertainties, the difference between the diel behavior and values of measured and modeled P($O_3$) is significant.

Upon further analysis of the discrepancy between measured and modeled P($O_3$) at high NO$_x$, it was found that the measured peroxy radical behavior as a function of NO was similar to studies previously reported in the literature. In these studies, the measured HO$_2$/OH ratio and measured RO$_2$ decrease less rapidly than that modeled for higher NO levels, causing measured

or calculated P($O_3$) to be several factors larger than modeled P($O_3$). As such, neither MOPS chamber artifacts nor reactive chlorine chemistry can fully explain this model-data P($O_3$) mismatch. While an additional HONO source proportional to NO$_x$ can help to improve diel P($O_3$) patterns, mid-morning HONO levels needed to approximate MOPS P($O_3$) are at least a factor of two higher than HONO levels observed in other environments, including ones nearby in Colorado. If an unknown RO$_2$ source proportional to NO is added to the model, we can approximate the measured P($O_3$) diurnal patterns within uncertainty levels.

However, this additional RO$_2$ does not fully explain the modeled and measured RO$_2$ disagreement at high NO. Therefore, if we can resolve measured and modeled peroxy radical disagreement and identify this unknown RO$_2$ source, then this additional radical source may be one solution to the model-data P($O_3$) mismatch.

More research must be conducted to understand the differences between modeled and measured $P(O_3)$. The second-generation MOPS is still in early stages of development and more rigorous testing is needed to decrease the MOPS absolute measurement uncertainty through the reduction of $O_3$ analyzer drifting and improvement in the precision of this analyzer. Conversely, model comparisons highlight the need to revisit current mechanism chemistry, including possible missing peroxy radicals at high NO or $NO_x$ levels.

If the MOPS accurately predicts morning $P(O_3)$, then $L_N/Q$ metrics from observation-constrained models that calculate radical mixing ratios may be incorrectly assessing $NO_x$ or VOC $O_3$ production sensitivity and the efficacy of $O_3$ reduction strategies. The use of these mechanisms in CTMs could create significant differences between modeled and observed $P(O_3)$ during peak $O_3$ production hours. Further, the plethora of chemical mechanisms available for use in these models create a large spread in model $O_3$ predictions. Thus, differences between measured and modeled $P(O_3)$ can have substantial and potentially costly implications for $O_3$ mitigation strategies that are put in place in $O_3$ NAAQS non-attainment areas. The MOPS measurements indicate that $P(O_3)$ in Golden, CO and its surrounding areas is more $NO_x$-sensitive than models currently predict in the morning hours, suggesting that $NO_x$ emission reductions in this region could be a viable solution for $O_3$ mitigation.

## 5  Data availability

The MCM version 3.3.1 is freely available at http://mcm.leeds.ac.uk/MCM/ and the University of Washington Chemical Model (UWCM) framework used to run MCMv331 is available to the public from G. Wolfe. Meteorological and chemical data collected during the DISCOVER-AQ and FRAPPÈ studies are available at http://www-air.larc.nasa.gov/missions/discover-aq/discover-aq.html and https://www2.acom.ucar.edu/frappe.

Author S. Brown serves on the editorial board of this journal. No other authors declare any conflicts of interest.

*Acknowledgements.* We gratefully acknowledge the entire DISCOVER-AQ and FRAPPÈ teams for the collection of ground and airborne measurement data in this work. We kindly acknowledge R. Cohen for the provision of data used in these model studies. PTR-ToF-MS measurements during DISCOVER-AQ were carried out by P. Eichler, T. Mikoviny, and M. Müller, and were supported by the Austrian Federal Ministry for Transport, Innovation and Technology (bmvit) through the Austrian Space Applications Programme (ASAP) of the Austrian Research Promotion Agency (FFG). We thank G. Wolfe for provision of and help with the MCM model framework, W. Goliff for the provision of the RACM2, and J. Thornton for thoughtful discussions. We thank the two anonymous reviewers for their useful comments. For the use of the web version of the HYSPLIT model (http://www.ready.noaa.gov), we acknowledge the NOAA Air Resources Laboratory. This work was funded by NASA grants NNX14AR83G and NNX12AB84G.

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

**Table 1.** Measured parameters input into the RACM2 and MCMv331. Inorganic chemical species measurement time resolution is 1 min. Aircraft chemical species were measured every 1 s. Evacuated whole-air canister VOC point measurements were interpolated to 1-h medians as described in Section 2.2. All measured constraints were either averaged or interpolated to 10 min for model runs.

| Number | Model input | Method[b] | Uncertainty (%) | Institution |
|---|---|---|---|---|
| 8 | **Inorganics** | | | |
| | $O_3$ | CL | 10 | EPA |
| | $SO_2$ | UV Fluorescence | 10 | |
| | $NO_2$, NO | CES/CAPS, CL | 10 | |
| | CO, $CO_2$, $CH_4$ | WACs/GC/GC-MS (Colman et al., 2001) | $\leq 5$ | UCI |
| | $HNO_3$ | TD-LIF (Day et al., 2002) | 25 | UC Berkeley[a] |
| 58 | **Organic Species** | | | |
| 42 | $C_2$-$C_{10}$ *NMHCs*, *organic nitrates*: | WACs/GC/GC-MS (Colman et al., 2001) | 3-100 | UCI |
| | ethane, ethene, acetylene, propane, propene, i-butane, n-butane, i-pentane, n-pentane, isoprene, n-hexane, n-heptane, n-octane, 2,3-dimethylbutane, 2-methylpentane, 3-methylpentane, 2,4-dimethylpentne, 2,2,4-trimethylpentane,cyclopentane,methylcyclopentane, cyclohexane, methylcyclohexane, benzene, toluene, ethylbenzene, m,p-xylene, o-xylene, 2-ethyltoluene, 3-ethyltoluene, 4-ethyltoluene, 1,3,5-trimethylbenzene, 1,2,4-trimethylbenzene, 1,2,3-trimethylbenzene, $\alpha$-pinene, $\beta$-pinene, methyl nitrate, ethyl nitrate, i-propylnitrate, 2-butylnitrate, 2-pentylnitrate, 3-pentylnitrate, 2-methyl-2-butylnitrate | | | |
| | *NMHCs[a]*: | PTR-ToF-MS (Müller et al., 2014) | 10 | U. Innsbruck |
| | methyl ethyl ketone, methanol, methyl vinyl ketone, methacrolein, acetic acid acetaldehyde, acetone | | | |
| | formaldehyde | DFGAS (Weibring et al., 2006, 2007) | 5 | CU-INSTAAR |
| | peroxy acetyl nitrate, peroxy propyl nitrate | PAN-CIMS (Zheng et al., 2011) | 13 | NCAR |
| | hydrogen peroxide, formic acid, acetic acid | PCIMS (Treadaway, 2015) | 30 | URI |
| | ethanol, d-limonene/3-carene, camphene | TOGA (Apel et al., 2003) | 30 | NCAR |

[a] Denotes aircraft measurements
[b] CL, chemiluminescence; CES, cavity enhanced spectroscopy; CAPS, cavity attenuated phase shift spectrometer; WAC, whole-air canister; GC, gas chromatography; GC-MS, gas chromatography mass spectrometer; TD-LIF, thermal dissociation laser-induced fluorescence; PTR-ToF-MS, proton transfer reaction time-of-flight mass spectrometer; DFGAS, difference frequency generation absorption spectrometer; CIMS, chemical ionization mass spectrometer ('PAN', peroxyacyl nitrate; 'P', peroxide); TOGA, trace organic gas analyzer.

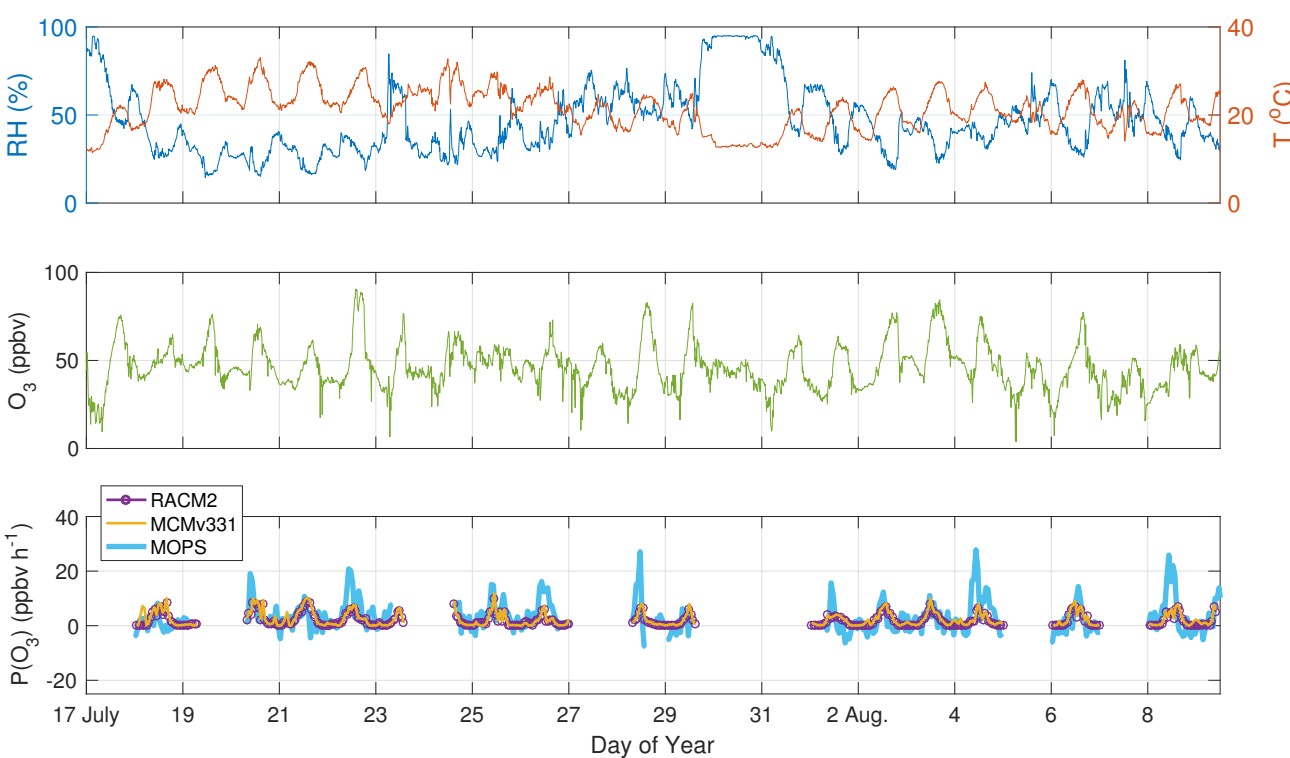

**Figure 1.** Top: Full-campaign 10-minute temperature and relative humidity in Golden, CO. The "warm" period is defined as days before 27 July 2014. Middle: Full-campaign 10-min $O_3$ mixing ratios for 17 July to 10 August 2014. Bottom: $P(O_3)$ measured by the MOPS and modeled from the RACM2 and MCMv331 for the same time period. Measured to modeled comparisons are shown for days with available MOPS measurements and are averaged over a 1-hour time period.

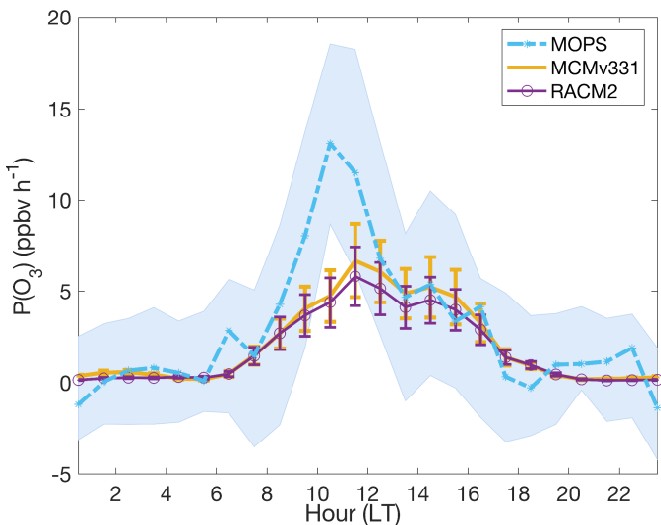

**Figure 2.** Full-campaign median hourly P(O$_3$) measured by the MOPS, and modeled by the RACM2 and MCMv331 for MOPS measurement days. Shaded areas represent the variance in MOPS P(O$_3$) due to the variation in the zero correction. The RACM2 and MCMv331 relative error bars are shown at the $1\sigma$ confidence level.

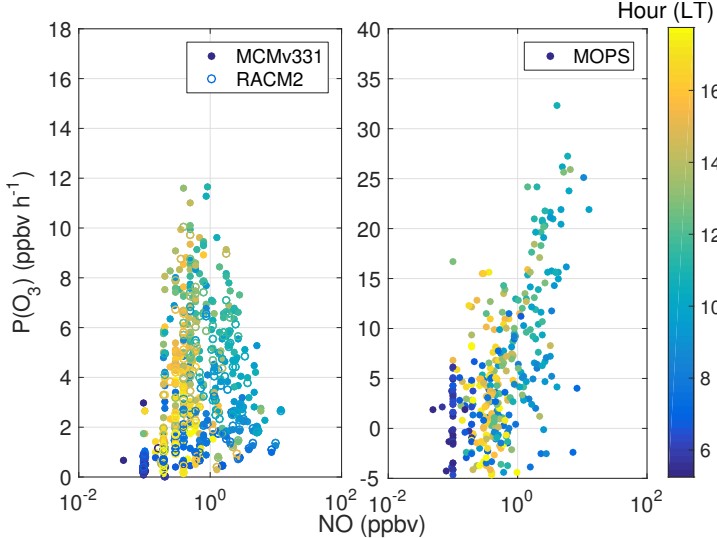

**Figure 3.** RACM2, MCMv331 (left), and MOPS (right) 30-minute P(O$_3$) as a function NO for all MOPS measurement days. Points are colored by hour of day from 0600-1800 LT.

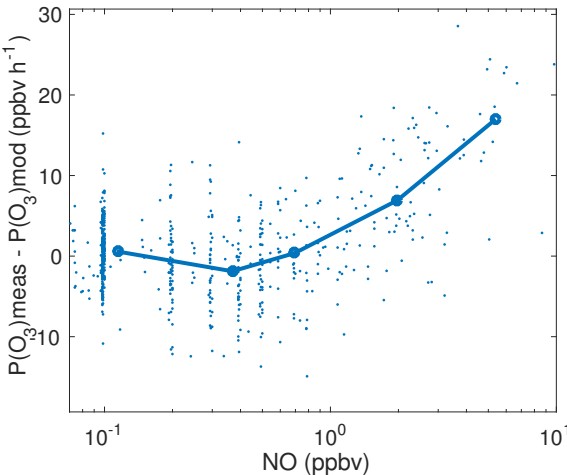

**Figure 4.** Difference between P(O$_3$) measured and modeled as a function of measured NO. Individual points are averaged for 30 minutes, while the solid line indicates the average P(O$_3$) difference binned by NO.

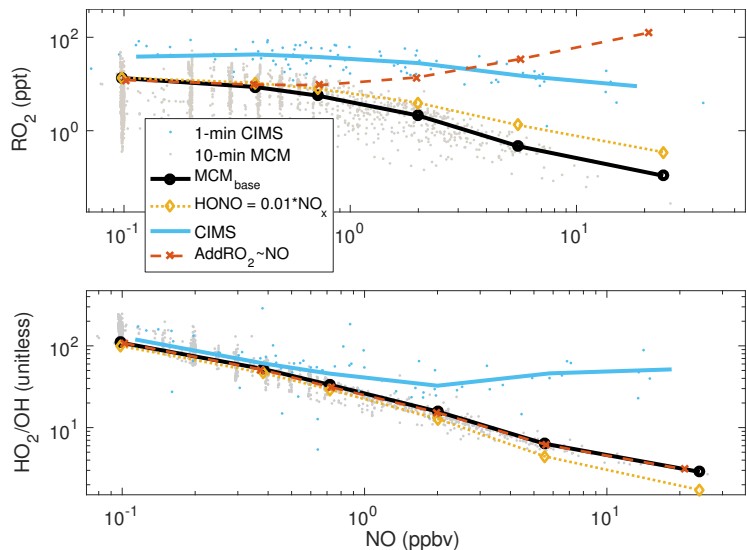

**Figure 5.** C-130 CIMS HO$_2$/OH ratio and RO$_2$ as a function of aircraft NO (chemiluminescence, 20 pptv $\pm$ 10%, 1$\sigma$ uncertainty) and modeled HO$_2$/OH ratio and RO$_2$ versus constrained NO measured continuously in Golden, CO. Aircraft measurements used are limited to the first 1 km in the boundary layer and for only times when the C-130 was within 20 km of Golden, CO. A well-mixed boundary layer is assumed for all measurements.

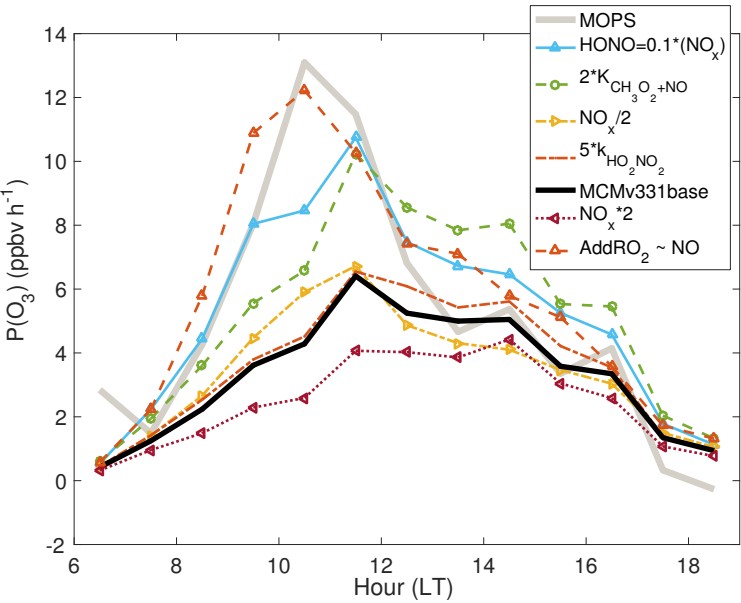

**Figure 6.** Model P(O₃) scenarios using MCMv331 calculated for daytime P(O₃) hours between 0600 and 1800 LT. Median hourly P(O₃) is derived from the model case studies described in the main text and compared to the MOPS median diel P(O₃) and MCMv331 median diel base case.

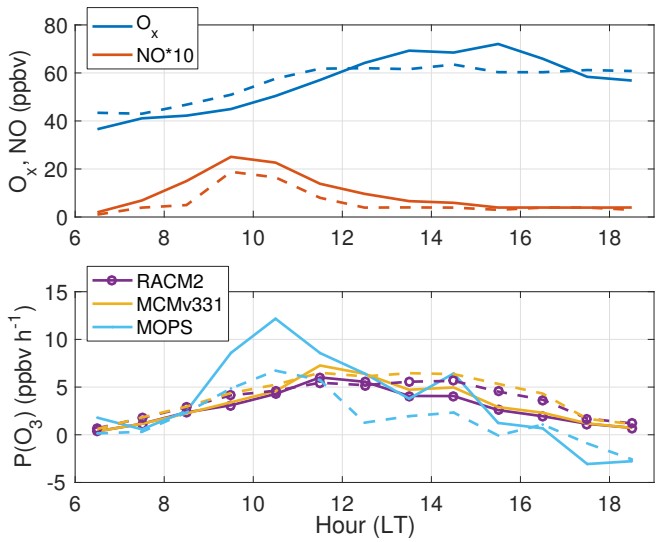

**Figure 7.** Top: $O_x$ ($O_3$ + $NO_2$) and NO mixing ratios for Denver plume (solid) versus all other days (dashed) from 17 July - 10 August 2014 in Golden, CO. Bottom: Median measured and modeled P(O₃) for Denver plume (solid) and non-Denver plume (dashed) days between 0600-1800 LT.