# Peer review of "Higher measured than modeled ozone production at increased $\mathbf{NO}_x$ levels in the Colorado Front Range"

_Atmospheric Chemistry and Physics, 2016_

## Referee Comment (RC1) · Anonymous Referee #2 · 18 Jan 2017

General comments

The manuscript describes a comparison of P(O3) measured using a second generation MOPS instrument and P(O3) modelled using a detailed chemistry mechanism (MCMv3.3.1) and lumped chemistry scheme (RACM2). A higher measured than modelled P(O3) is reported during the morning hours and the authors postulate that this discrepancy could be caused by a model under-estimation of HO2 under high NOx conditions and present a possible reaction which cycles OH to HO2 via NO that could rectify the discrepancy. The manuscript concludes with a discussion on how these model uncertainties may influence ozone reduction strategies in the region. From reading this manuscript in detail, I have not been convinced that the differences reported between

modelled and measured P(O3) are not, in part, caused by instrumental issues (I have detailed my specific concerns below) and to a smaller extent, caused by uncertainties in the model constraints (e.g. the interpolation of the VOC measurements and the use of a constant, median VOC value for certain VOCs). I cannot recommend publication in ACPD before the specific comments I have raised below are adequately addressed.

Specific comments

Throughout the manuscript, the authors argue that the higher observed P(O3) than modelled P(O3) at high NO is supported by observations of higher (than modelled) HO2 under high NO conditions. Only recently, however, have HO2 measurements been reported in the literature that have not been influenced by artefact signals induced by the fast decomposition (within instruments) of certain RO2 species. The authors do not discuss recent observations and model comparisons made in China (Tan et al., 2017) where interference-free HO2 observations agree well with model predictions under elevated NO conditions. The authors need to provide a more balanced discussion of modelled and measured HO2 comparisons under high NO conditions which includes a review of the possible problems relating to HO2 measurements as well as the more recent HO2 modelled and measured comparisons.

Pg 5, line 8 -16: The authors discuss the influence of humidity on the O3 analyser used in the MOPS instrument and cite previous papers which have reported large changes in O3 concentration as RH is rapidly changed. How fast and by what percentage did RH typically change during these ambient measurements made in Golden? Did the authors attempt to correct for this RH dependence?

Pg 5, line 23: 'some excess HONO was measured..' how much and how did this vary with RH, T, [NO2] and actinic flux? Would this excess HONO be expected to be greatest during the morning hours when NO2 concentrations were high? Details like this need to be provided so the differences reported between modelled and measured P(O3) during the morning hours can be fairly assessed.

Pg 5, line 24: '..actions were taken'. Some details on the actions that were taken should be provided here.

Pg 6, discussion of model constraints: I have some concerns over how well the model constraints represent the local conditions in Golden. How many WAC samples were collected between 0700 – 1200? I worry that interpolating these canister samples may lead to underestimations in peak VOC concentrations. In table S1 the authors state that constant, median values of various species measured on board the aircrafts were used to constrain the model. How can this be a valid approach given the high variability of some of these species, e.g. the biogenics? Would this variability bias the model predictions as a function of time of day? I.e. were VOC concentrations highest during the morning when the boundary layer was lowest? It is unclear if the sensitivity analysis conducted (presented in Fig S1) included varying VOC concentrations by the observed variability or varying by the uncertainty in the VOC measurement only?

Pg 7, line 1: '..integrated for 24 hours.' I would be surprised if the reactive intermediates and, more importantly, modelled $HO_2$ and $RO_2$ are in steady state after 24 hours. By what percentage did the modelled peroxy radical concentrations change from day 1 to day 2?

Pg 8, line 19: How does the diel measured $P(O_3)$ change if the negative $P(O_3)$ points are not omitted? By omitting these negative points, does this not positively biased the $P(O_3)$ measurements in the morning when these negative points are most frequent? Were the modelled $P(O_3)$ calculated at the same time also removed? Could the authors be more quantitative when they discuss the $O_3$ analyser RH dependence – what do they class as a 'drastic change' in RH?

Section 3.2 would benefit from Rate of Production Analysis. This would help to reveal what is driving the high modelled $P(O_3)$ on certain days. It would also reveal the cause of the differences that were observed between RACM and MCM at times rather than rely on speculation '..perhaps due to more explicit treatment of VOCs..'

Pg 9, lines 6-16: The authors give some reasons as to why they have confidence in the MOPS measurements during the morning hours when the model and measurements do not agree. I am not convinced, however, that the MOPS measurements are not positively biased at this time: By removing the negative P(O3) values which most frequently occurred during the morning hours, I would expect this to positively bias the measurements at this time even if a median rather than the mean value is used. Furthermore, I would expect heterogeneous HONO production to be greatest when NO2 concentrations are high, and, this would also artificially positively bias the measurements during the morning most.

Pg 10, lines 1-5: The measurement bias reported due to photolytic HONO production is of the same magnitude as the discrepancy between model and measured P(O3) observed during the morning (Fig 2). Although the authors argue that individual modelled and measured P(O3) deviations are often larger than the bias caused by photolytic HONO production, there seems to be, from studying figure 1, a similar number of positive and negative deviations of the MOPS P(O3) from the modelled P(O3) suggesting that many of these deviations are caused by noise rather than missing or uncertain model chemistry.

Section 3.3.4: A rate of 9 -15x10-11 cm3 molecule-1 s-1 for the postulated reaction 'OH+NO (+O2) → HO2 + NO2' cannot be justified given that the known reaction rate for OH reacting with NO is two to three times slower. The impact of this highly speculative reaction is presented as positively influencing the modelled P(O3) in line with the MOPS observations - this is misleading, however: Figure S2 highlights that although inclusion of this reaction, proceeding at a rate of 9 -15x10-11 cm3 molecule-1 s-1, improves the modelled and measured agreement in the morning, it actually worsens the agreement between the modelled and measured P(O3) during the afternoon; there is no comment about this in the text, however. How does this reaction influence other modelled species, e.g. OH?

Section 3.3.6: Some discussion of the modelled HONO sources is needed here. If only

gas-phase sources, i.e. OH+NO are considered, the model likely under-estimates the actual HONO that was present.

Reference Tan et al., Atmos. Chem. Phys., 17, 663-690, 2017

---

## Referee Comment (RC2) · Anonymous Referee #1 · 27 Jan 2017

Comments on B. Baier et al. Higher measured than modeled ozone production at increased NOx levels in the Colorado Front Range

Overall this paper is interesting but not entirely convincing. However, it is the authors staking their reputation on this and I will not stand in their way. However, I hope that the many co-authors are sufficiently convinced that they will not also agree to be co-authors any paper using their data that comes to a different conclusion about high NOx photochemistry.

There are three major issues the authors should address:

1) They dismiss large negative values but not large positive values as unphysical. It is

not clear that both types of errors shouldn't be treated as representative of the random error of the method used.

2) Much is made of the linear NOx dependence of the model observation mismatch. A figure showing this should be presented.

3) The authors dismiss HONO and VOCR as a potential source of discrepancies too easily. It would be better to instead assume the entire discrepancy is due to HONO or to VOCR that is correlated with NO and show the reader the relationships between NO, HONO and VOCR that would be required to explain the results. Then the reader can judge whether those ideas are plausible.

Other comments

Page 2 Line 27: Only the formation of peroxyacyl nitrates, and not alkyl nitrates, is a form of Ox loss. The reaction RO2 + NO -> RONO2 does not produce or consumer either O3 or NO2.

Page 2 Line 27: Equation 3 includes the production of nitric acid via OH + NO2 - this should also be included in the description of ozone chemical loss processes.

Page 4 Line 1- 3: It would be useful to understand which of these references are from ambient field studies and which (if any) are from chamber studies. Some of the proposed explanations of the MOPS v. Modeled PO3 in this paper should exist in both field and chamber studies (e.g. OH + NO2 + O2) while others (e.g. ClNO2) are unlikely to be a factor in chamber studies.

Page 4 Line 13 - 20: It is unclear what this paragraph is actually describing. Are these results all from a single study, or are these similar results from a variety of MOPS deployments?

Page 8, Line 22: Were time periods selected for removal based on the humidity or of the MOPS measurement itself? It is unclear from the description in this paragraph and should be clarified.

Page 9, Line 17-19: While Figure 1 shows several days where the MOPS peak PO3 value is between 10-20 ppb/hr greater than the modeled value, it also shows several days where the MOPS minimum PO3 value is between -10 and -20 ppb/hr (e.g. Jul 21, 23, 24, Aug 6). Since these large negative values are explained as non-physical, why is it not equally plausible that the large positive values are a measurement artifact as well?

Page 11 Line 12 & Page 12 Line 5: The claim that the difference between MOPS and Modeled PO3 is linear with NO is difficult to evaluate from the figures provided. Since the NO dependence of the MOPS-Model disagreement is an important part of the analysis, the authors should include a figure that directly shows the difference in PO3 during the morning as a function of NO.

Page 12, Line 8-9: This sentence appears to contradict section 2.2, where median diurnal values of VOCs are said to be used in the model calculation. The authors should clarify how the model includes the dependence of VOC reactivity on NO.

Page 12, Line 15-17: As written, the proposed OH + NO + O2 -> HO2 + NO2 reaction seems to be an OH loss as well as an HO2 source. The authors should clarify why the argument here against a missing VOC source do not also apply to this proposed reaction.

Page 12, Line 27-Page 13, Line 5: I am uncertain whether the mechanisms listed in this paragraph are candidates for the OH + NO + O2 reaction mechanism or not. Rewording this paragraph to clarify what some possible forms of this reaction are, and whether it might involve additional species such as acetylene would be useful.

Section 3.3.6: Given the known difficulties models have in capturing HONO concentrations, this section feels overly brief in dismissing HONO as a HOx source. I would appreciate at least brief consideration of what magnitude HONO source would be necessary to make the modeled PO3 values match the MOPS values.

[Figure]

Figure 5: If the disagreement between measured and modeled PO3 is strictly a function of NOx, why is the disagreement so much worse on plume days, even though plume and non-plume days both have average morning NO concentrations of approximately 2 ppb? Are there other significant differences between plume and non-plume days, or is there significantly more variability in the amount of NOx on plume than on non-plume days?

Technical Corrections Page 12, Line 12: If this number is a rate of HO2 production, then the units on this are incorrect (perhaps intended to be radicals cm-3 s-1?) Page 13, Line 1-2: In this sentence, the phrase "the formation of" appears to be extraneous. The accent on FRAPPÉ is inconsistent. Sometimes the campaign is incorrectly written as FRAPPÈ.

---

## Author Comment (AC1) · 19 May 2017

Please find our comments (non-italicized) to the following author comments (italics) addressed below:

*Overall this paper is interesting but not entirely convincing. However, it is the authors staking their reputation on this and I will not stand in their way. However, I hope that the many co-authors are sufficiently convinced that they will not also agree to be coauthors any paper using their data that comes to a different conclusion about high NOx photochemistry. There are three major issues the authors should address:*

We thank the reviewer for his or her time spent on providing a review for this manuscript. We agree that the results presented here are not entirely convincing and have conducted further laboratory tests, improved our data analysis, done more model runs, and re-written the manuscript to focus on the evidence for ozone production rates and its implications.

We will ignore the reviewer's second sentence – perhaps the reviewer should be gently reprimanded by the editor for it? The third sentence is interesting in that it implies that doing science is staking a position and sticking to it, even if previous evidence is found to be wrong or is overwhelmed by the weight of new evidence. In our view, science advances as evidence accumulates and scientists should change their thinking in light of new evidence, even if it contradicts their previous conclusions. So we respectfully disagree with the reviewer on these two sentences.

On the other hand, the reviewer makes many good points in the rest of the review and we will do our best to answer them.

*1) They dismiss large negative values but not large positive values as unphysical. It is not clear that both types of errors shouldn't be treated as representative of the random error of the method used.*

As pointed out in Baier et al. (2015), this technique seems to produce a large negative differential ozone signal that is roughly correlated with temperature, relative humidity, or actinic flux. This negative ozone differential is not related to negative chemical ozone production because it occurs even when the reference chamber film has been removed from the reference chamber (so that ozone production is the same in both chambers) and on low ozone production days with low levels of actinic flux and precursor gases. It should be noted that the raw $P(O_3)$ data show positive deviations from this negative pattern between 0800-1200 LT on most days. Thus, this large negative is related to something occurring in the chambers or in the Thermo ozone monitor.

After we received the reviews, we investigated in the laboratory that sensitivity of the Thermo 49C ozone monitor to relative humidity and to temperature. We did not find a strong correlation between differences in temperature between the sample and the reference chamber, but we did find a rather large shift in the Thermo instrument zero as a function of relative humidity equivalent to roughly 70% ambient relative humidity. The shift is almost binary, with an offset that is steady up to within a few percent of a certain relative humidity and then a sudden shift of ~2 ppbv within a few percent around that value. Thus, we restrict our analysis to days with lower relative humidity, when we are confident that a relative humidity initiated ozone offset change did not happen.

Four zeros were applied to the raw $P(O_3)$ data during the campaign. Two were by removing the ultem (reference chamber) film for an entire 24-hour period and two were using known low ozone production days as zeros. To avoid the problems with the sudden shift in the instrument zero, we have restricted our analysis to days when the ambient relative humidity was below 70%. In addition, we use zeros only from days that have similar relative humidity values and variations as the days for which the ozone production rate is being measured.

We include a figure of the resulting $P(O_3)$ offset that we subtracted from the observed $P(O_3)$ measurement in the Supplemental Information Figure S1 in order to get the corrected $P(O_3)$ that is shown in subsequent figures within the manuscript.

In order to ensure that the reported $P(O_3)$ values were not sensitive to our choices, we varied the threshold relative humidity and the choice of zero days that we used in the analysis over wide ranges. While the peak ozone production rate varied from 8 to 15 ppbv/hr and the minimum early evening values varied from 2 to -4 ppbv/hr, the overall shape of the measured ozone production rate was the same (i.e. MOPS corrected $P(O_3)$ data increase around 0800 LT, peak around 9-11 LT, and then decrease later in the day. We added a figure in the supplemental material to show this (Figure S2).

We used a Thermo 49C instrument to make these measurements, and at the level of the expected ozone difference from chemical ozone production (i.e. an ozone differential of 0.3 ppbv is equivalent to 8 ppbv/hr), the data are noisy and need to be integrated over time to reduce the noise. In Figure 1 that shows the measured values as a function of day of the year, we chose a one-hour average as opposed to the 30-minute averages chosen in the original manuscript submission. It is clear that the negatives are no longer as prominent – mainly because of the improved zero correction and partly because of the increased integration time, but the positive feature at ~1000 LT remains quite robust.

Additional information has been added on Pg. 6 lines 4-22 outlining these comments above, as well as in Section 1 in the Supplemental Information. We have also added the following lines to section 3.2 Pg. 11 lines 6-11:

"Third, when different relative humidity thresholds are used to correct the raw $P(O_3)$ data, measured $P(O_3)$ consistently exhibits the same diurnal behavior with a positive deviation from modeled $P(O_3)$ around 1000 LT. Fourth, deviations from the $O_3$ differential baseline derived from zeroing methods are observed between 0900-1100 LT even before correcting the MOPS measurements. Thus, we have confidence in the positive MOPS $P(O_3)$ signatures, which are greater than the modeled $P(O_3)$ during the morning hours."

*2) Much is made of the linear NOx dependence of the model observation mismatch. A figure showing this should be presented.*

We should correct ourselves in that the model-data mismatch increases monotonically for increasing NO above 1 ppbv. A figure showing the NO dependence of the model observation mismatch now been added to the manuscript (now Figure 4) along with the following text on page 10 line 31 to page 11 lines 1-2:

"The missing modeled $P(O_3)$ appears to be monotonically increasing with NO for NO values greater than roughly 1 ppbv (Fig. 4). The difference between measured and modeled $P(O_3)$ is near zero up to 1 ppbv NO and almost 15 ppbv/h at 5 ppbv NO. This unexpected increase in $P(O_3)$ with increasing NO provides a clue as to what might be causing the difference between measured and modeled $P(O_3)$ ."

*3) The authors dismiss HONO and VOCR as a potential source of discrepancies too easily. It would be better to instead assume the entire discrepancy is due to HONO or to VOCR that is correlated with NO and show the reader the relationships between NO, HONO and VOCR that would be required to explain the results. Then the reader can judge whether those ideas are plausible.*

This is a good suggestion. We have added to Sections 3.3.3 and 3.3.4, which use model calculations to assess whether HONO or $RO_2$ sources could potentially explain the model-data mismatch during the morning hours. We have further described the effect that these two additional $P(O_3)$ sources have on

the $HO_2$/OH ratio as a function of NO, but have not been able to reproduce the measured $HO_2$/OH $NO_x$ dependence. (Please see the following comments below that discuss the plausibility of HONO as a morning $P(O_3)$ source.)

An additional $HO_2$ source via the generic reaction $RO_2$ + NO -> $HO_2$ + $NO_2$ + R'O was added to the MCMv331 using $CH_3O_2$ as the organic peroxy radical. However, the addition of this reaction elevates the entire modeled $P(O_3)$ curve throughout the day and does not alter morning $P(O_3)$ patterns relative to the MOPS. Additional $RO_2$ added to the MCMv331 proportional to approximately 0.05% $NO_x$ was added to the MCMv331, which more closely replicates the MOPS morning $P(O_3)$, but underestimates the MOPS afternoon $P(O_3)$. Please see Figure 6, which was taken from the supplemental material and added to the main text. We have added text beginning on Pg. 13 line 28 to Pg. 14 lines 1-15 to discuss the plausibility of missing model $RO_2$/VOCR.

*Other comments*

*Page 2 Line 27: Only the formation of peroxyacyl nitrates, and not alkyl nitrates, is a form of Ox loss. The reaction RO2 + NO -> RONO2 does not produce or consumer either O3 or NO2.*

Fixed in Eq. 3. Pg. 2 lines 26 – Pg. 3 line 1 now read:

"Equation 3 describes the chemical $O_3$ (or $NO_2$) destruction rate or rate of removal to reservoir species as the fraction of O1D molecules resulting from $O_3$ photolysis that react with $H_2O$ to form OH; reactions of $O_3$ with $HO_x$ ($HO_2$ + OH); the net production of peroxyacyl nitrates (PANs); the reaction of OH and $NO_2$ to form nitric acid ($HNO_3$) and $O_3$ loss through reactions with alkenes and halogens."

*Page 2 Line 27: Equation 3 includes the production of nitric acid via OH + NO2 - this should also be included in the description of ozone chemical loss processes.*

Fixed in Eq. 3 and on Pg. 2 lines 26- Pg. 3 line 1.

*Page 4 Line 1- 3: It would be useful to understand which of these references are from ambient field studies and which (if any) are from chamber studies. Some of the proposed explanations of the MOPS v. Modeled PO3 in this paper should exist in both field and chamber studies (e.g. OH + NO2 + O2) while others (e.g. ClNO2) are unlikely to be a factor in chamber studies.*

All of the studies listed in the last paragraph on Pg. 4 lines 5-7 are field studies. To our knowledge, no chamber studies have been published that explain the measured vs. modeled $P(O_3)$ discrepancy or the $HO_2$/OH vs. NO discrepancy also mentioned. Thus, to our knowledge, there have not been any chamber studies conducted that could propose an explanation to the results shown in this manuscript. The proposed reaction (OH + $NO_2$ + $O_2$) has been removed as a possible solution to the observed model-data mismatch.

*Page 4 Line 13 - 20: It is unclear what this paragraph is actually describing. Are these results all from a single study, or are these similar results from a variety of MOPS deployments?*

We clarify this sentence to state which deployments (and when) were MOPS deployments, and to note that all of the other studies referenced were field studies. Pg. 4 lines 28-30 now read:

"Due to the aforementioned discrepancies between measured and modeled radicals, $P(O_3)$ calculated from measured peroxy radicals can routinely be more than double the $P(O_3)$ calculated from modeled $HO_2$ or $RO_2$ at high $NO_x$ according to several field studies conducted during the past decade… "

Page 4 Lines 33-37 now read:

"For example, in 2010 the first version of the MOPS (Cazorla et al., 2012, Ren et al., 2013 compared directly-measured $O_3$ production rates to both modeled $P(O_3)$ and that calculated from measured peroxy radicals. Observed $P(O_3)$ was approximately equal to that modeled for NO levels up to 1 ppbv, but was significantly larger for higher values of NO."

*Page 8, Line 22: Were time periods selected for removal based on the humidity or of the MOPS measurement itself? It is unclear from the description in this paragraph and should be clarified.*

Please read the description of the analysis method given earlier in our responses.

*Page 9, Line 17-19: While Figure 1 shows several days where the MOPS peak PO3 value is between 10-20 ppb/hr greater than the modeled value, it also shows several days where the MOPS minimum PO3 value is between -10 and -20 ppb/hr (e.g. Jul 21, 23, 24, Aug 6). Since these large negative values are explained as non-physical, why is it not equally plausible that the large positive values are a measurement artifact as well?*

Please read the description of the analysis method given earlier in our responses. Many of these large negative deviations have been reconciled with a) averaging to 1-hour time periods and b) through revising the correction of the raw $P(O_3)$ data to account for only days when the relative humidity was less than 70%.

*Page 11 Line 12 & Page 12 Line 5: The claim that the difference between MOPS and Modeled PO3 is linear with NO is difficult to evaluate from the figures provided. Since the NO dependence of the MOPS-Model disagreement is an important part of the analysis, the authors should include a figure that directly shows the difference in PO3 during the morning as a function of NO.*

Please see our comment addressing this above. Figure 4 has been added in the text.

*Page 12, Line 8-9: This sentence appears to contradict section 2.2, where median diurnal values of VOCs are said to be used in the model calculation. The authors should clarify how the model includes the dependence of VOC reactivity on NO.*

We should clarify that the model includes the average VOC reactivity dependence on NO. VOC reactivity is linked to the median diurnal values of the VOCs used to constrain the model. Thus, the whole-air sampled VOCs are the available sources providing an average dependence of VOC reactivity on NO in the chemical mechanisms used in this study. We have stated this on Pg. 13 lines 24-26:

"Although measurements of VOC reactivity were not available during the field campaign time period and thus are unavailable for comparison to modeled VOC reactivity, the suite of VOCs measured in Golden should sufficiently capture the average VOC reactivity dependence on NO in the mechanisms used here."

*Page 12, Line 15-17: As written, the proposed OH + NO + O2 -> HO2 + NO2 reaction seems to be an OH loss as well as an HO2 source. The authors should clarify why the argument here against a missing VOC source do not also apply to this proposed reaction.*

This proposed reaction has been removed from the analysis, as preliminary laboratory testing and theoretical work has shown that this reaction is highly unlikely. Please see our comment above about a missing VOC source.

*Page 12, Line 27-Page 13, Line 5: I am uncertain whether the mechanisms listed in this paragraph are candidates for the OH + NO + O2 reaction mechanism or not. Rewording this paragraph to clarify what some possible forms of this reaction are, and whether it might involve additional species such as acetylene would be useful.*

This paragraph has been omitted.

*Section 3.3.6: Given the known difficulties models have in capturing HONO concentrations, this section feels overly brief in dismissing HONO as a HOx source. I would appreciate at least brief consideration of what magnitude HONO source would be necessary to make the modeled PO3 values match the MOPS values.*

Section 3.3.4 has been thoroughly modified on Pg. 14 starting on line 16. We have conducted additional model tests in order to describe the magnitude of a HONO source that would allow the modeled $P(O_3)$ and MOPS $P(O_3)$ to match. We have also described model sources of ambient HONO here. An additional HONO source is one probable missing element from our model analyses. As HONO production should be linearly dependent upon $NO_x$ levels, additional HONO is added to MCMv331 as 10% of $NO_x$ values, yielding peak HONO levels at 1000 LT of 0.9 ppbv. While the modeled $P(O_3)$ with added HONO best replicates the MOPS diel $P(O_3)$ pattern shown in Figure 6 out of all of the model case studies conducted for this analysis, HONO levels needed to do so are almost twice levels observed at a nearby observation site in Colorado as well as in other, more polluted areas.  Modeled peak $P(O_3)$ is shifted to approximately an hour later than the MOPS and is slightly overestimated in the afternoon. We have added the following text on Pg. 15  lines 3-10 to indicate this:

"A HONO source proportional to $NO_x$ was added to the MCMv331, resulting in average HONO levels of 0.5-0.9 ppbv between 0700 and 1200 LT, with peak HONO levels of 0.9 ppbv at 1000 LT when MOPS $P(O_3)$ exhibits its diel peak. This case study approximately replicates the observed morning $P(O_3)$ and observed OH within uncertainty levels (Fig. 6). However, while added HONO in the MCMv331 improves the agreement between observed and modeled diel $P(O_3)$ ,mid-morning HONO levels needed to do so are over a factor of two higher than those observed in other areas within Colorado (Brown et al., 2013 Vandenboer et al., 2013) and in environments with much higher $NO_x$ levels (Vandenboer et al., 2015). Thus, the $HO_2$/OH ratio and the abnormally high HONO required to match the observed $P(O_3)$ provide evidence that at most only a part of the observed $P(O_3)$ can be explained by atmospheric HONO. "

*Figure 5: If the disagreement between measured and modeled PO3 is strictly a function of NOx, why is the disagreement so much worse on plume days, even though plume and non-plume days both have average morning NO concentrations of approximately 2 ppb? Are there other significant differences between plume and non-plume days, or is there significantly more variability in the amount of NOx on plume than on non-plume days?*

The difference between measured and modeled $P(O_3)$ is dependent upon $NO_x$. We have shown that the chemical mechanisms may be missing a hydroperoxy radical source co-emitted with NO relative to measurements and have explored several model case studies to try to reconcile this model-measurement disagreement.

 It is noted in Section 2.2 Pg. 7 lines 27-30 that $NO_x$ and VOCs are both higher, on average, on plume days than on non-plume days. We have also shown that there can be significant differences (~5-10

ppbv/h) between the modeled and MOPS P(O$_3$) at NO levels greater than approximately 1 ppbv. Thus, the difference between the MOPS and models on plume vs. non-plume days is consistent with this result.

We have added the following text to Pg. 17 lines 10-12:

 "Measured O$_x$ maxima are 2-7 ppbv greater on these ``plume'' days than when air is advected from elsewhere, and higher P(O$_3$) is measured by the MOPS than is modeled by the RACM2 and MCMv331 (Fig. 7). This result is roughly consistent with the difference between measured and modeled P(O$_3$) as a function of NO shown in Fig. 4."

*Technical Corrections Page 12, Line 12: If this number is a rate of HO2 production, then the units on this are incorrect (perhaps intended to be radicals cm-3 s-1?)*

Corrected.

*Page 13, Line 1-2: In this sentence, the phrase "the formation of" appears to be extraneous. The accent on FRAPPÉ is inconsistent. Sometimes the campaign is incorrectly written as FRAPPÈ.*

Corrected.

---

## Author Comment (AC2) · 19 May 2017

Please find our comments (non-italicized) to the following author comments (italics) addressed below:

*The manuscript describes a comparison of P(O3) measured using a second generation MOPS instrument and P(O3) modelled using a detailed chemistry mechanism (MCMv3.3.1) and lumped chemistry scheme (RACM2). A higher measured than modelled P(O3) is reported during the morning hours and the authors postulate that this discrepancy could be caused by a model under-estimation of HO2 under high NOx conditions and present a possible reaction which cycles OH to HO2 via NO that could rectify the discrepancy. The manuscript concludes with a discussion on how these model uncertainties may influence ozone reduction strategies in the region. From reading this manuscript in detail, I have not been convinced that the differences reported between modelled and measured P(O3) are not, in part, caused by instrumental issues (I have detailed my specific concerns below) and to a smaller extent, caused by uncertainties in the model constraints (e.g. the interpolation of the VOC measurements and the use of a constant, median VOC value for certain VOCs). I cannot recommend publication in ACPD before the specific comments I have raised below are adequately addressed.*

We thank the reviewer for presenting several important points to be addressed in the manuscript. We note that we have adjusted the manuscript focus to address peroxy radical ($HO_2$ and $RO_2$) discrepancies at high $NO_x$ and have performed additional model case studies exploring the plausibility of missing peroxy radical sources. Since we proposed the reaction OH + NO (+ $O_2$) to form $HO_2$ + $NO_2$ earlier in 2016, preliminary work in our laboratory and that of a colleague, as well as a theoretical study, have all indicated that this reaction is unlikely. We have thus decided to omit this reaction as an explanation to the model-data mismatch described in the main text. We have instead explored the plausibility of missing peroxy radicals and other chemical causes.

*Throughout the manuscript, the authors argue that the higher observed P(O3) than modelled P(O3) at high NO is supported by observations of higher (than modelled) HO2 under high NO conditions. Only recently, however, have HO2 measurements been reported in the literature that have not been influenced by artefact signals induced by the fast decomposition (within instruments) of certain RO2 species. The authors do not discuss recent observations and model comparisons made in China (Tan et al., 2017) where interference-free HO2 observations agree well with model predictions under elevated NO conditions. The authors need to provide a more balanced discussion of modelled and measured HO2 comparisons under high NO conditions which includes a review of the possible problems relating to HO2 measurements as well as the more recent HO2 modelled and measured comparisons.*

The introduction has been modified to include a) known $HO_2$ measurement interferences and b) an extended literature review of $HO_2$ measurement comparisons with models including more recent studies. This extension of the introduction includes a synthesis of comparisons made in high and low-NO environments in various studies (Stone et al. (2012)) as well as other studies

such as Tan et al. (2017) and Hofzumahaus et al. (2009), which show small and/or insignificant model $HO_2$ underprediction during morning hours. However, Tan et al. (2017) $HO_2$ measurements do not suffer from the $RO_2$ interference and yet still exhibits slight underprediction of $HO_2$ at high $NO_x$. The highest NO values reported for some of these studies was only a few ppbv as in Tan et al., 2017. For many studies, the separation between measured and modeled $HO_2$ is within uncertainties at these low NO levels but then grows at higher NO levels, particularly above 10 ppbv. Thus, even when accounting for $HO_2$ interferences, model underestimation of $HO_2$ at high $NO_x$ appears to be robust. If the ratio of atmospheric $RO_2$ to $HO_2$ is used to predict a maximum $HO_2$ interference, this interference can be approximately a factor of two higher than $HO_2$. Thus, an $HO_2$ interference is not the sole cause for model-data $HO_2$ mismatch at high $NO_x$ presented in the literature studies outlined in the introduction. These ideas have also been added to Pg. 3 lines 1-25.

We have also modified the introduction to describe model $RO_2$ underestimation, presenting this as a viable reason for model $P(O_3)$ underprediction. Model peroxy radical underestimation at high NO is therefore outlined throughout the rest of the paper as a possible cause for model underestimation of $P(O_3)$ and case studies are presented in the revised manuscript.

*Pg 5, line 8 -16: The authors discuss the influence of humidity on the O3 analyser used in the MOPS instrument and cite previous papers which have reported large changes in O3 concentration as RH is rapidly changed. How fast and by what percentage did RH typically change during these ambient measurements made in Golden? Did the authors attempt to correct for this RH dependence?*

Additional laboratory studies have been conducted to address this comment. Please see our comments for Reviewer 1 that address this issue.

*Pg 5, line 23: 'some excess HONO was measured..' how much and how did this vary with RH, T, [NO2] and actinic flux? Would this excess HONO be expected to be greatest during the morning hours when NO2 concentrations were high? Details like this need to be provided so the differences reported between modelled and measured P(O3) during the morning hours can be fairly assessed.*

We would expect the HONO bias to be greatest during the morning hours when $NO_x$, relative humidity, and actinic flux is high. However, we would expect that the $P(O_3)$ bias due to excess HONO production in Golden, CO be considerably smaller as we have been able to decrease this bias ~ 30% in State College, PA (Baier et al. 2015) by covering the MOPS inlet face to avoid excess HONO production and photolysis, and as $NO_x$ levels Golden, CO are considerably smaller than in Houston, TX. The relative humidity and $NO_x$ levels in Golden, CO were significantly lower than in Houston, TX, while the actinic flux is similar.

Model calculations of the MOPS HONO bias using the measured chamber HONO to observed $NO_x$ ratio in Houston, TX and scaling this ratio to the observed $NO_x$ in Golden, CO indicate a bias of 3 ppbv/h. However, in this estimation, we did not consider that only a small fraction of

chamber-generated HONO would be photolyzed within the residence time in the camber, which makes this potential positive P(O₃) interference less than 0.5 ppbv/h and thus insignificant. We expect that, since this bias is considerably smaller than the difference between measured and modeled P(O₃) shown in Figure 1, the model-measurement P(O₃) mismatch is not due to this artifact.

We have added lines Pg. 6 line 32 to Pg. 7 line 5 describing what we have done to decrease the HONO bias. We have added how we have calculated this potential HONO bias for Golden, CO on Pg. 11 line 30 to Pg. 12 line 8.

*Pg 5, line 24: '..actions were taken'. Some details on the actions that were taken should be provided here.*

Please see previous comment.

*Pg 6, discussion of model constraints: I have some concerns over how well the model constraints represent the local conditions in Golden. How many WAC samples were collected between 0700 – 1200? I worry that interpolating these canister samples may lead to underestimations in peak VOC concentrations.*

A total of 46 whole-air canister samples were taken between 0700-1200 LT. Therefore, over 64% of the WAC samples were taken during the morning hours. As continuous VOC measurements were not available in Golden, CO, it is not possible to determine median diurnal values of these VOCs in any other manner. We have added the following text to lines 18-24 on Pg. 7:
"In the absence of continuous ground-based VOC measurements, $C_2$-$C_{10}$ non-methane hydrocarbons (NMHC) and organic nitrates were measured from 72 total whole-air canister (WAC) samples that were collected in Golden and analyzed by gas chromatography (GC) and gas chromatography-mass spectrometry (GC-MS) in the laboratory. An average of five samples were taken daily over 16 days, with approximately 64\% of sampling occurring between 0700 and 1200 local time (LT) to capture VOC mixing ratios during morning $O_3$ production hours, and sparser sampling in the afternoon between 1400 and 1800 LT to examine advection from sources east of Golden, CO such as the Denver metropolitan and Commerce City regions."

In addition, Shirley et al. (2006) employ a similar approach to estimating VOC abundances, but use OH reactivity measurements to calculate VOC speciation abundances. Given several model sensitivity runs varying these VOCs, we expect that the estimation of VOC abundances in this manuscript can adequately capture diurnal variations -- in an average sense -- in order to compare hourly measured and modeled P(O₃) rates in Figure 2.

*In table S1 the authors state that constant, median values of various species measured on board the aircrafts were used to constrain the model. How can this be a valid approach given the high variability of some of these species, e.g. the biogenics? Would this variability bias the model predictions as a function of time of day? I.e. were VOC concentrations highest*

*during the morning when the boundary layer was lowest? It is unclear if the sensitivity analysis conducted (presented in Fig S1) included varying VOC concentrations by the observed variability or varying by the uncertainty in the VOC measurement only?*

Aircraft measurements in the boundary layer for areas surrounding Golden, CO were available after 9am LT for the P-3B and after 10am LT for the C-130. Thus, the concentrations of aircraft VOCs used in this analysis are limited to mainly after 9-10am LT and early morning concentrations were not known. Because of this limited aircraft data, the few morning VOC species that were measured on aircraft are set at constant, median values, while the majority of VOCs are taken directly from whole-air canister samples as described above. For those chemical species measured in common between the aircraft and the ground site, the agreement is within 30% on average. Further, biogenic chemical species do not have a substantial role in OH reactivity in Golden and so would have a smaller impact on modeled $P(O_3)$ Thus, the most important chemical species measurements for ozone production came from the whole air sampler on the ground. We have added Pg. 7 line 33 to Pg. 8 line 5 to clarify the few aircraft points that we have available to incorporate into the model:

"Canister VOCs were supplemented by boundary layer inorganic and organic chemical species measurements obtained on the NASA P-3B and NSF/NCAR C-130 aircraft and constant, median values were calculated for the limited times of the day when these aircraft were in the vicinity of Golden and used in the model (Table 1, Table S1). Aircraft measurements for Golden were available after 0900 LT on P-3B overflights, which occurred up to three times daily, while C-130 measurements were available after 1000 LT when this aircraft was within roughly 20 km of the measurement site. Airborne measurements of inorganic and organic species agree to within 30% on average."

The analysis in Table S1 is conducted by varying the VOC concentrations by their uncertainty only in order to determine the total model uncertainty that is due to parameter uncertainties as in Chen et al (2010). This analysis was conducted to solely investigate the dependence of model uncertainty on model parameter uncertainties.

*Pg 7, line 1: '..integrated for 24 hours..' I would be surprised if the reactive intermediates and, more importantly, modelled HO2 and RO2 are in steady state after 24 hours.*
*By what percentage did the modelled peroxy radical concentrations change from day 1 to day 2?*

Additional MCMv331 model runs have been conducted that vary the integration time from one to three days with little difference seen between model $HO_2$, $RO_2$, or OH for these three different model integration times. Therefore, we do not expect these species to vary significantly whether we use a one-, two-, or three-day integration period for model runs. We have further explained this result on P8 lines 16-18 and deem a 24-hour integration period for these species acceptable:

"In addition, a one-day integration time is calculated to be sufficient for radical concentrations

and intermediate species to reach steady-state as a two-fold or even three-fold increase in this integration time period does not impact radical concentrations or the P(O₃) results described below."

*Pg 8, line 19: How does the diel measured P(O3) change if the negative P(O3) points are not omitted? By omitting these negative points, does this not positively biased the P(O3) measurements in the morning when these negative points are most frequent? Were the modelled P(O3) calculated at the same time also removed? Could the authors be more quantitative when they discuss the O3 analyser RH dependence – what do they class as a 'drastic change' in RH?*

Please see our comments for Reviewer 1 on this issue.

*Section 3.2 would benefit from Rate of Production Analysis. This would help to reveal what is driving the high modelled P(O3) on certain days. It would also reveal the cause of the differences that were observed between RACM and MCM at times rather than rely on speculation '..perhaps due to more explicit treatment of VOCs..'*

RACM2 and MCMv331 produce essentially the same P(O₃) for all but a few short time periods. This model difference is far from the focus of this paper. Therefore, we have dropped the discussion of these differences and their possible cause in order to keep the focus on the differences between the measured and modeled P(O₃) which is unaffected by these infrequent RACM2 – MCMv331 differences.

*Pg 9, lines 6-16: The authors give some reasons as to why they have confidence in the MOPS measurements during the morning hours when the model and measurements do not agree. I am not convinced, however, that the MOPS measurements are not positively biased at this time: By removing the negative P(O3) values which most frequently occurred during the morning hours, I would expect this to positively bias the measurements at this time even if a median rather than the mean value is used. Furthermore, I would expect heterogeneous HONO production to be greatest when NO2 concentrations are high, and, this would also artificially positively bias the measurements during the morning most.*

Please see comments for Reviewer 1 describing new correction techniques and providing evidence for the MOPS measurements being real in the mid-morning. Our additional analyses show that the MOPS diurnal pattern is rather robust for whichever correction to the raw P(O₃) data that we apply.

Please also see our comments above as to why we believe that HONO production (whether gas-phase or heterogeneous) is not a major issue in the MOPS chambers causing the measured P(O₃) to deviate from modeled P(O₃) in the morning. Deviations of P(O₃) from the average zero correction are seen in the MOPS raw data as stated in lines 15-17 on Pg. 6. These deviations from the MOPS raw data are greater than +3 ppbv/h: our calculated HONO bias from model

simulations. If we account for the amount of HONO that could possibly be photolyzed during the chamber residence time, this bias drops to less than 0.5 ppbv/h as mentioned above.

*Pg 10, lines 1-5: The measurement bias reported due to photolytic HONO production is of the same magnitude as the discrepancy between model and measured P(O3) observed during the morning (Fig 2). Although the authors argue that individual modelled and measured P(O3) deviations are often larger than the bias caused by photolytic HONO production, there seems to be, from studying figure 1, a similar number of positive and negative deviations of the MOPS P(O3) from the modelled P(O3) suggesting that many of these deviations are caused by noise rather than missing or uncertain model chemistry.*

Please see our comments for Reviewer 1 regarding the noise level on the MOPS data. By limiting the MOPS measurement time periods to those with ambient relative humidity below 70%, a majority of the observed $P(O_3)$ noise is reduced. In addition, we average the MOPS data in Figure 1 in the main text to one hour. With this averaging, noisy data is also reduced and deviations from the hourly MOPS $P(O_3)$ still exhibit differences relative to modeled $P(O_3)$ that are larger than the HONO biases we estimate.

*Section 3.3.4: A rate of 9 -15x10-11 cm3 molecule-1 s-1 for the postulated reaction 'OH+NO (+O2) > HO2 + NO2' cannot be justified given that the known reaction rate for OH reacting with NO is two to three times slower. The impact of this highly speculative reaction is presented as positively influencing the modelled P(O3) in line with the MOPS observations - this is misleading, however: Figure S2 highlights that although inclusion of this reaction, proceeding at a rate of 9 -15x10-11 cm3 molecule-1 s-1, improves the modelled and measured agreement in the morning, it actually worsens the agreement between the modelled and measured P(O3) during the afternoon; there is no comment about this in the text, however. How does this reaction influence other modelled species, e.g. OH?*

The reviewer is correct that the formation rate of the OH+NO adduct is known, but it has not been measured in atmospheric pressure air. It is possible that $O_2$ could act in some way to speed up this reaction. However, preliminary laboratory studies by us and another group and theoretical work have indicated that this reaction is unlikely to be fast enough even in the presence of oxygen. We have omitted this discussion as a possible explanation for the model-data mismatch discussed in this manuscript.

*Section 3.3.6: Some discussion of the modelled HONO sources is needed here. If only gas-phase sources, i.e. OH+NO are considered, the model likely under-estimates the actual HONO that was present.*

It is possible for the model to underestimate HONO. However, studies that have used measured HONO to constrain the models HONO still cannot resolve the $HO_2$ discrepancy at high $NO_x$, providing some evidence that a further $HO_2$ source at high $NO_x$ may still be needed to resolve

this issue.  HONO levels needed to match the MOPS P(O$_3$) are a factor of two too high as discussed in comments to Reviewer 1.

We have significantly modified section 3.3.4 on Pgs. 14-15 to discuss model HONO studies and sources of HONO present in the models used here.

*Reference Tan et al., Atmos. Chem. Phys., 17, 663-690, 2017*

This reference has been added to Pg. 4 in the Introduction.

---

## Author Response (AR2)

Author response to Reviewer #1 (reviewer comments in italics):

*The paper is much improved. I recommend publication. The authors might consider whether there are other observable consequences of their ideas. For example, is the lifetime of any short lived VOC likely to be lower during the high PO3 period--and therefore to have a different relationship with NO? Or have daughter molecules that have rapidly changing concentrations? I don't think adding that is necessary to publication but I find myself curious what other observable consequences there might be.*

We thank the anonymous reviewer for their comments.

The lifetime of short-lived VOCs is governed largely by its reaction rate with OH and the ambient OH concentration.  In this work, we have not found measured OH to be substantially different from that modeled and so it is hypothesized that chemical species' lifetimes will not be shortened. We could imagine that high amounts of NO could shift the VOC degradation pathways to chemical products that result in more HO2 but these pathways would have to be ones that are not already contained in RACM2 or MCMv331. This idea is certainly worth exploring, however, since we have not yet been able to identify a viable mechanism for producing this excess HO2 nor to explain these P(O3) observations.

We have explored in Section 3.4 perhaps one of the most direct consequences for the results found in this manuscript both from an atmospheric chemistry and ozone mitigation standpoint: that higher measured morning P(O3) may shift morning NOx-VOC sensitivity towards NOx-sensitive regimes relative to models, and that it has the potential to alter calculations of ozone advection using the ozone budget equation.

Author response to Reviewer #2 (reviewer comments in italics):

*Introduction, pg 4, lines 15 – 35: I found this revised section a little misleading. I think this could be rectified if the magnitudes of modelled to measured HO2 (and modelled to measured p(O3)) disagreement are compared in absolute terms. In the Tan et al (2017) study, modelled and measured HO2 agree within uncertainty at 6 ppbv NO. By eye, (but the authors can provide an accurate number) at 6 ppbv NO in this study P(O3) measured is ~20 ppb hr-1 greater than modelled. The Tan et al (2017) study does not support the magnitude of the P(O3) discrepancy reported in this work and this needs to be reflected in the Introduction.*

We thank this anonymous reviewer for their helpful comments.

We have examined Tan et al. (2017) and think that we should focus on Figure 9 in Tan et al (2017), which is reproduced here. We do not see any data for NO = 6 ppbv, so we will focus on the data at NO = 4 ppbv.

[Figure]

We have altered Pg. 4 lines 16-18 to say:

"Removing this interference therefore improves model-measurement HO$_2$ agreement at high NO$_x$ within uncertainty levels in some studies, especially for NO less than 10 ppbv (Tan et al., 2017; Shirley et al., 2006) but cannot fully explain model HO$_2$ under-predictions at high NO$_x$ in others (Griffith et al., 2016; Brune et al., 2015)."

We respectfully disagree with the second part of this comment. The average difference between the MOPS and modeled P(O3) at 4 ppbv NO is approximately +17 ppbv/h. The median difference between the Tan et al (2017) calculated minus modeled P(O3) at 4 ppbv NO is >+40 ppbv/h, thus supporting the magnitude of the P(O3) discrepancy reported in this work given uncertainty levels.

*The authors have made some attempts to improve the corrections applied to the raw MOPS data by limiting times when RH was below 70%. It is not obvious from looking at the latest time-series in Figure 1 that the MOPS data coverage presented has reduced. Was this threshold RH already applied before the initial submission?*

This threshold was not applied to the initial submission. The MOPS data coverage presented in Fig. 1 has indeed been reduced since the first submitted draft of this manuscript. Thus, tiny gaps in the data have been removed from the original Fig. 1 when the MOPS Thermo analyzer lines exceeds a relative humidity of 70%. However, since we have increased our averaging time interval to one hour, we have likely averaged over these gaps to create the plot shown in the third panel in Fig. 1, which is why the new filter used is not obvious from looking at the latest time series.

*There is still a lot of uncertainty associated with all the corrections applied, however. The authors state that the negative ozone production rates are 'roughly correlated with temperature, relative humidity, or actinic flux'. Could the authors provide an equation of the form:*

*Artificial signal = Temperature × a(±b) + RH × c(±d) + hv × e(±f)*
*This will provide some confidence in the corrections applied*

We agree with the reviewer that being able to write such an equation would give more confidence in the correction for the negative P(O3) signal. We have conducted many tens of hours of studies trying to ascertain the cause of this negative signal and to quantify its relationships with temperature, relative humidity and actinic flux. However, we have found no such simple relationship. The three or four other groups that have developed ozone production rate sensors have had no more success than we have had. It may be that there are important parameters that we have not considered. This problem is clearly an interesting photochemistry problem, probably heterogeneous, but it is beyond the scope of this paper.

We have changed the words "that are roughly correlated" on Pg. 5 line 31 to "appear to be roughly correlated" as no laboratory tests had validated this statement for the Baier et al. (2015) study. Temperature and relative humidity inside of the MOPS ozone analyzer are related. However, we state on Pg. 6 lines 5-8 that laboratory testing has shown that temperature differences do not significantly affect the artificial ozone signal in the ozone analyzer, though relative humidity does significantly affect the ozone drifting.

*Related to this point, the authors discuss the days chosen for the zero measurements: 'zeros that were taken only on days with diurnal patterns and absolute values of relative humidity that are similar to most MOPS measurements days'. The authors need to be specific here, what do they mean by 'similar' and 'most'? Furthermore, this statement begs the question, how similar was temperature and actinic flux on these days also given the correlation with these parameters too?*

We note that days chosen to represent "zero" measurements are ones with absolute values of relative humidity and diurnal patterns similar to all other campaign measurement days. That is, the diurnal patterns show a decrease in relative humidity from the morning to afternoon hours and an increase in relative humidity as nighttime approaches. Days where a zero was taken, but rain showers or thunderstorms occurred during the day have been disregarded because the ambient humidity likely worsens the ozone drift, providing a drift correction that is likely dissimilar to drifting on all other measurement days. Temperature and actinic flux for zero days were within the range of the temperature and actinic flux on zero days were within the range of that measured on days when we were not zeroing the MOPS instrument but again, we only consider relative humidity patterns as this most significantly affects the MOPS ozone drifting. We have clarified Pg. 6 lines 20-24 to state:

"Zeros that were taken only on days with diurnal patterns and absolute values of relative humidity that reflect the range of relative humidity measured on non-zeroing, MOPS measurements days were used in this analysis. Days chosen for zeroing the MOPS instrument with elevated ambient relative humidity compared to non-zeroing days likely have enhanced ozone analyzer drifting, according to our laboratory simulations of field operations. Thus, these days will not provide an average zero correction to the MOPS data because they are anomalous."

*It is not clear in Figure S2 which line relates to which RH?*

As we calculated the MOPS analyzer relative humidity using an estimated temperature of the ozone analyzer environment and the ambient vapor pressure, the MOPS analyzer relative humidity is continuously calculated. Figure S2 and its caption have now been corrected to the following:

[Figure]

Figure S 2: Variation in the 24-hour corrected $P(O_3)$ by incrementally varying the MOPS analyzer temperature (and thus, relative humidity) filter. MOPS analyzer average relative humidities ranging from 50-80% for each temperature scenario from 21°C (corresponding to 47% mean RH) to 12°C. The MOPS corrected $P(O_3)$ shown is averaged for 30 minutes.

to show the mean RH for each temperature scenario, with temperatures ranging from 21C to 12C. In addition, we have added further text in Section S1 explaining this figure in more detail, along with clarifying that the relative humidity to which we are referring is that of the MOPS ozone analyzer:

"Although we note that artificial positive and negative P(O3) can be correlated with temperature, differences in temperature between sample and reference chamber did not play a large role in initiating baseline drifting. However, the MOPS O3 analyzer did exhibit large baseline shifts greater than 2 ppbv when air enters the analyzer at relative humidities greater than 70%. Due to this relative humidity dependent baseline drifting, the MOPS raw P(O3) data correction techniques are adjusted from Baier et al. (2015) so as to minimize MOPS measurement days when O3 analyzer drifting was more severe. Because the MOPS ozone analyzer is sheltered in an environment that is air conditioned to temperatures below ambient values, the MOPS ozone analyzer relative humidity does exceed 70% based on laboratory calculations using an expected MOPS analyzer environment temperature and the ambient vapor pressure. Thus, MOPS data are filtered to times when the air entering the ozone analyzer has a relative humidity below 70%. Furthermore, zeros that were taken only on days with diurnal patterns and absolute values of relative humidity within the range of relative humidities measured on non-zeroing, MOPS measurements days were used to correct the raw P(O3) data. These time periods should capture the average P(O3) baseline drift throughout the campaign. Due to this zero filtering, one zero was discarded leaving three to be used to correct the raw P(O3) data. The average zero correction that is subtracted from the raw P(O3) measurements in order to derive the corrected P(O3) is shown in Fig. S1.

We have further tested the robustness of this threshold using a wide range of analyzer relative humidities to ensure that our corrected P(O3) values were not sensitive to this threshold choice. This testing was done by varying the temperature of the MOPS ozone analyzer to replicate field conditions. Figure S2 shows the half-hourly median MOPS P(O3) diurnal signal that results from varying the analyzer relative humidity."

*Section 3.3 is much improved from the initial submission, however, a couple of modifications are needed: Pg 12, line 32: 'either all of these measurements methods contain similar artifacts..' this argument does not hold. Why, if this is an instrumental problem rather than missing chemistry does this have to relate to a similar artifact? Rather, the MOPS artificial signal may derive from heterogeneous production of radicals or radical precursors whilst the previous FAGE measurements could be influenced by an RO2 interference. This sentence needs to be modified to reflect this.*

We have clarified Pg. 11 lines 20-23 to reflect this important point, stating:
"The MOPS observations, independent of these studies, yield similar results for the dependence of P(O3) on NO indicating that the MOPS and other measurement methods both contain artifacts that act to increase P(O3) in a similar manner, or that the model-measurement disagreement occurs due to differences in the chemistry between observational and computational methods used to determine O3 production rates."

However, radical measurements were made in the first version of the MOPS, showing similar ambient and chamber radical concentrations. Therefore, we do not expect that heterogeneous production of radicals interferes with the MOPS measurement.

*The authors argue that the positive excursion in P(O3) is apparent even before the zero correction is applied and use this as evidence that the positive excursion is not an instrumental artefact (pg 12, lines 22 – 27). I do not agree with this reasoning. It may be true that the zero correction is not inducing this positive P(O3) observed in the mrornings, but other artifacts, for example, artifacts induced by*

*heterogeneous NO2 reactions in the chamber could be contributing to this signal. The authors mention on Pg 7, line 25 'excess HONO of up to five times ambient values was measured in the MOPS chambers' and that the 'production mechanism has not been identified' Really, systematic laboratory studies are needed to determine the absolute magnitude of this heterogeneous HONO production as a function of NOx. Failing that, the authors should at least modify the discussion in section 3.3.1 to reflect the fact that the model measurement discrepancy at high NOx could be caused by a currently unidentified HONO (or radical) production mechanisms in the MOPS that may not necessarily scale with NOx as assumed in section 3.3.1.*

We now clarify that we assume a certain HONO production mechanism to calculate an estimated HONO bias for Golden, CO. However, continuing with this assumption, we also state that even if the HONO was produced by a different mechanism, levels would have to be an order of magnitude larger than our estimated bias to contribute significantly to the model-measurement P(O3) discrepancy and furthermore, levels of HONO that are an order of magnitude higher have not been measured in the MOPS chambers, as stated in Baier et al. (2015). Thus, we deem this scenario unlikely to be a significant cause for this discrepancy, although we acknowledge that it can be a small contributor. Pg. 12 lines 3-18 now read:

"Additionally, adsorbed NO2 can result in heterogeneous formation of HONO, and a HONO source within the chambers may result in excess P(O3) from artificial OH production (Baier et al., 2015). For the Golden, CO study, NOx levels were a factor of three lower on average than in Houston, TX, the relative humidity was 35% lower on average, and the actinic flux was similar. Although identifying MOPS chamber HONO production mechanisms will require more intensive laboratory studies, we assume that the largest HONO source within the MOPS chambers stems from NO2 adsorption on the chamber walls. Thus, NOx levels in Golden can be used to infer MOPS chamber HONO levels in Golden. We have applied the observed chamber HONO:NOx ratio in Houston, TX to the Golden, CO study because, under this assumption, HONO production should depend linearly on NOx adhering to the walls. We have calculated a maximum diurnal bias of +3 ppbv/h at 1000 LT (2 sigma) that decreases later in the day to less than 1 ppbv/h as NOx decreases. However, this calculated P(O3) bias is rather conservative; the chamber residence time of 130 s and the HONO photolysis frequency for Golden, CO can be used to determine the percentage of chamber HONO that would be converted into O3-producing radicals. In doing so, less than 15% of chamber HONO is photolyzed. Consequently, the bias for Golden, CO would be less than 0.5 ppbv/h and would contribute insignificantly to the observed P(O3) signal. In order to explain observed and modeled P(O3) differences in Golden by chamber-induced HONO production, HONO levels would need to be more than an order of magnitude larger. Given the levels of MOPS chamber HONO measured in Houston and other areas (Baier et al., 2015), the likelihood of excess chamber HONO production being a significant cause for the resultant differences between modeled and measured P(O3) is small.

*Figure 6, Some comment is needed about why P(O3) in the MCM3.3.1 RO2=0.0005x[NOx] scenario is lower (despite [RO2] concentrations being higher) than the base MCM case in the afternoon.*

The result before in Figure 6 indicated a model run constraining RO2 (as CH3O2) levels to be proportional to NOx instead of adding an RO2 source proportional to NOx. As the MOPS P(O3) indicates that there could be unknown RO2 sources in the models, we have run a new MCMv331 scenario that adds an unknown RO2 proportional to NO (as this species is likely co-emitted with NO) to match the P(O3) discrepancy in addition to the RO2 sources produced from the VOCs we input into the models. The [RO2] from this model run is in better agreement with the base case [RO2], only slightly

overestimating the MOPS afternoon P(O3) signal.  Because of this model result, we have changed Section 3.3.3 Pg 13 line 32 to Pg. 14 line 17 to read:

"One hypothesis is that a missing HO2 or RO2 source linearly scalable to NO was not included in the models. However, forty-two total C2-C10 VOCs were measured by whole-air canister samples representing a large suite of organic chemical species within the models including ones with high OH reactivities that are particularly important for O3 formation. We have tested two hypotheses: first, that an RO2 source that reacts with NO to form HO2 and NO2 is missing in the models despite the suite of VOC measurements made in Golden, and second, that an unknown peroxy radical source co-emitted with NO can explain the missing modeled P(O3).

To test our first hypothesis, we have added a generic reaction involving RO2 and NO to form additional HO2 + NO2 + RO in the MCMv331, enhancing the HO2 produced from RO2 + NO reactions. Since the reaction between the methylperoxy radical (CH3O2) and NO is the dominant organic peroxy radical reacting with NO to form new O3 in both chemical mechanisms, this species' reaction rate coefficient was used for this model test. By essentially doubling the CH3O2 rate constant, this reaction only elevates modeled P(O3) throughout the day, does not alter the diurnal P(O3) pattern (Fig. 6), and does not resolve the discrepancy between measured and modeled peroxy radicals at high NO.

To test our second hypothesis, additional RO2 was added to the model in order to match the peak morning diel MOPS P(O3) in Fig. 6 and was scaled to NO as this unknown species is likely co-emitted with NO. This addition improves model-measurement P(O3) agreement in the morning and only slightly overestimates the afternoon diel P(O3) relative to the MOPS, agreeing with measured P(O3) within uncertainty levels. In magnitude, the model-measurement RO2 agreement is improved with this case study. However, the RO2 continues to increase while measured RO2 decreases at higher NO (Fig. 5). Thus, this model case study suggests that an unknown missing RO2 source could possibly explain the differences between measured and modeled P(O3) if this discrepancy between measured and modeled RO2 can be resolved and if the identity of the unknown RO2 can be found."

This new result slightly changes our conclusions to state that an unknown RO2 source co-emitted with NO is plausible. However, as we measure a full suite of VOCs including ones most significant in O3 formation, we are not able to identify this peroxy radical source. We have changed our conclusions slightly to read on Pg. 18 lines 22-32:

"Upon further analysis of the discrepancy between measured and modeled P(O3) at high NOx, it was found that the measured peroxy radical behavior as a function of NO was similar to studies previously reported in the literature. In these studies, the measured HO2/OH ratio and measured RO2 decrease less rapidly than that modeled for higher NO levels, causing measured or calculated P(O3) to be several factors larger than modeled P(O3). As such, neither MOPS chamber artifacts nor reactive chlorine chemistry can fully explain this model-data P(O3) mismatch. While an additional HONO source proportional to NOx can help to improve diel P(O3) patterns, mid-morning HONO levels needed to approximate MOPS P(O3) are at least a factor of two higher than HONO levels observed in other environments, including ones nearby in Colorado. If an unknown RO2 source proportional to NO is added to the model, we can approximate measured P(O3) morning and afternoon diurnal patterns within uncertainty levels. However, this RO2 addition does not fully explain modeled and measured RO2 disagreement at high NO. Therefore, if we can resolve measured and modeled RO2 disagreement and identify this unknown RO2 source, then this additional peroxy radical source may be one solution to the model-data P(O3) mismatch.",

as well as Pg. 1 in the abstract lines 10-13:

"The P(O3) and peroxy radical (HO2 and RO2) discrepancy observed here is similar to those presented in prior studies. While a missing atmospheric organic peroxy radical source from volatile organic compounds co-emitted with NO could be one plausible solution to the P(O3) discrepancy, such a source has not been identified and does not fully explain the peroxy radical model-data mismatch. "

*Also the scenario where RO2+NO > HO2 + NO2 is not shown in Figure 6 as indicated in lines 27, 28, pg 16 (unless it is represented by the green dashed line, in which case the legend needs correcting).*

We have corrected the error in the legend of Fig. 6. Here, "XO2" has been replaced with "2*K_CH3O2+NO" but was, indeed, the green dashed line.

[revised manuscript text omitted]